# Olaph: Improving Factuality in Biomedical Long-form Question Answering

## Abstract

In the medical domain, numerous scenarios necessitate the long-form generation ability of large language models (LLMs). Specifically, when addressing patients' questions, it is essential that the model's response conveys factual claims, highlighting the need for an automated method to evaluate those claims. Thus, we introduce **MedLFQA**, a benchmark dataset reconstructed using long-form question-answering datasets related to the biomedical domain. We use MedLFQA to facilitate a cost-effective automatic evaluations of factuality. We also propose **Olaph**, a simple and efficient framework that utilizes cost-effective and multifaceted automatic evaluation to construct a synthetic preference set and answers questions in our preferred manner. Our framework leads us to train LLMs step-by-step to reduce hallucinations and include crucial medical claims. We highlight that, even on evaluation metrics not used during training, LLMs trained with our Olaph framework demonstrate significant performance improvement in factuality. Our findings reveal that a 7B LLM trained with our Olaph framework can provide long answers comparable to the medical experts' answers in terms of factuality. We believe that our work could shed light on gauging the long-text generation ability of LLMs in the medical domain. Our code and datasets are available.

## 1 Introduction

With the increasing versatility and exceptional performance of large language models (LLMs), their utilization in the medical or clinical domain is expanding rapidly (Singhal et al., 2023; Chen et al., 2023; Thirunavukarasu et al., 2023; Sun et al., 2024; Tu et al., 2024; Labrak et al., 2024; Jeong et al., 2024). One of the greatest advantages of LLMs in these domains is their capability to assist or even replace physicians' tasks (Egli, 2023; Tian et al., 2024). This includes scenarios such as question answering (multi-choice (Jin et al., 2021; Hendrycks et al., 2020; Jin et al., 2019; Pal et al., 2022; Xiong et al., 2024) or span-based (Krithara et al., 2023)), reporting a patient's Electronic Health Record (Thirunavukarasu et al., 2023; Yang et al., 2022), and conversations based on patient inquiries (Liévin et al., 2024). In the medical domain, numerous situations necessitate the long-form text-generation ability of LLMs. Among these, answering questions posed by patients demands conveying factual and crucial claims, highlighting the necessity for an automated method to evaluate these responses.

To address this challenge, it is important to measure the ability of open-foundation LLMs to answer in a long-form text. Thus, we aim to verify it through long-form question-answering (LFQA) tasks. LFQA is a task that requires elaborate and in-depth answers to open-ended questions (Fan et al., 2019; Stelmakh et al., 2022). Here, two main challenging points arise: One is that models should not hallucinate or generate false information (Min et al., 2023; Wei et al., 2024; Manes et al., 2024). For example, when a patient asks, *what could be causing the white tongue?* the response should convey crucial information about why white tongue occurs and its causes (e.g., *white tongue is usually caused by a buildup of bacteria and dead cells on the surface of the tongue*) while ensuring that incorrect information (e.g., *it is usually harmful and permanent*) is not provided.

Another challenge lies in the difficulty of automatically evaluating long-text responses. Existing tasks such as summarization or LFQA assess whether appropriate words are used and the semantic meaning is well encapsulated (Min et al., 2023; Falke et al., 2019; Laban et al., 2022; Fabbri et al., 2022; Krishna et al., 2023). Furthermore, other methods consist of manually verifying the

Table 1: Statistics of long-form question answering benchmark datasets containing patients' questions, answers, and two statements. We use an abbreviation for question (Q), answer (A), must-have statements (MH), and nice-to-have statements (NH) respectively. Texts highlighted in **bold** are generated using GPT-4 API calls. Some of the questions are filtered due to the ambiguous points.

| Dataset | Format (Original → Modified) | # of QA pairs | # of Ambiguous Questions | Avg. Length of Answers | Avg. # of MH statements | Avg. # of NH Statements |
|---|---|---|---|---|---|---|
| LiveQA (Abacha et al., 2017) | (Q, A) → (Q, A, **MH**, **NH**) | 100 | 4 | 82.8 | 2.9 | 3.0 |
| MedicationQA (Abacha et al., 2019) | (Q, A) → (Q, A, **MH**, **NH**) | 666 | 24 | 55.5 | 2.6 | 2.3 |
| HealthSearchQA (Singhal et al., 2023) | (Q) → (Q, **A**, **MH**, **NH**) | 3,077 | 96 | 118.8 | 2.6 | 2.3 |
| K-QA Golden (Manes et al., 2024) | (Q, A, MH, NH) | 201 | 1 | 88.5 | 4.4 | 3.5 |
| K-QA Silver (Manes et al., 2024) | (Q) → (Q, **A**, **MH**, **NH**) | 904 | 106 | 99.9 | 2.4 | 2.0 |

responses generated by LLMs using human annotators to ensure high factuality and absence of hallucination which are cost-ineffective and labor-intensive (Liu et al., 2023b; Fu et al., 2023; Liu et al., 2023a). In particular, in the medical field, it's also important to ensure that the information provided is accurate, up-to-date, and comprehensible to practitioners and patients alike. Developing reliable automatic evaluation methods would greatly enhance the efficiency and scalability of these assessments, leading to rapid and extensive advancements in the research field by reducing reliance on human evaluators.

To this end, we aim to gather existing LFQA datasets and reconstruct them as a benchmark for the automatic evaluation of medical responses. **MedLFQA** allows evaluating an LLM's response in detail: whether they effectively include the keywords necessary to answer the question, whether they are semantically similar to the answer, and whether they accurately include crucial claims without delivering hallucinated information. Furthermore, we employ GPT-4 (OpenAI, 2023b) to generate long-form answers and statements if needed (Section 3.1). For validation, we assess the answers and statements through three medical experts for pairwise evaluation. Thus, we identify that GPT-4 generated responses are reliable to use as the MedLFQA benchmark (Section 3.2).

We then introduce a simple and efficient framework **OLAPH** (**O**ptimizing **L**arge language models' **A**nswers with **P**references of mitigating **H**allucination), which leverages cost-effective and automatic evaluation to generate synthetic preference sets that can help align the model with preferred responses. Our OLAPH framework is composed of iterative learning through preference optimization on the synthetic preference sets. We first leverage supervised fine-tuning (SFT) to tailor a pre-trained LLM to a question-answering task (Ouyang et al., 2022) (Section 4.1). Then, we derive $k$ sampled predictions using temperature sampling (Guo et al., 2017) to construct synthetic preference set by adopting cost-effective and multifaceted automatic evaluations (Section 4.2). Then, we construct a preference set in every steps using previous-step models with self-generated responses and iteratively train with alignment tuning until convergence (Section 4.3 and 4.4). Overall, our framework generates synthetic preference sets using automatic evaluation metrics and iteratively trains LLMs with preferred responses the model generates.

Our findings reveal that learning through our OLAPH framework step-by-step can enhance long-text generation abilities by prioritizing factuality, semantic similarities, and word composition. We find that making a synthetic preference set with self-generated responses based on a wide range of evaluation criteria and iteratively training on the set increases the desired abilities in a long-text generation. Our findings also highlight that, even on evaluation metrics not used during training, LLMs equipped with our OLAPH framework demonstrate significant performance improvement in factuality. Surprisingly, 7B models trained with our framework can generate long-form answers comparable to medical experts' answers which are proven to be high-quality answers.

Overall, our contributions are as follows: (1) We introduce **MedLFQA**, a benchmark dataset with restructured formats of current biomedical LFQA benchmark datasets that enables automatic evaluation of the long-text generation ability of open foundation LLMs. (2) In this process, we constitute two statements that can automatically evaluate factuality cost-effectively through medical claims originated by long answers, aiding in a comprehensive understanding of long-text generation abilities. (3) We introduce the simple and efficient **OLAPH** framework, which leverages automatic evaluation to generate synthetic preference sets and employs iterative learning through preference optimization. (4) In our findings, we demonstrate that 7B models can generate long answers comparable to the medical experts' answers in terms of factuality.

## 2 PRELIMINARIES

### 2.1 LONG-FORM QUESTION ANSWERING

Long-form question answering (LFQA) is a task requiring elaborate and in-depth answers to open-ended questions (Fan et al., 2019; Stelmakh et al., 2022; Krishna et al., 2021). In the biomedical and clinical domains, LFQA is essential for effectively integrating AI into real-world applications. Despite its importance, there has been relatively little effort to construct patient-centered LFQA datasets due to its domain specificity. In other words, numerous scenarios necessitate the long-text generation ability of LLMs in these domains but provided with restricted amounts of usable data due to removing the identifying details for privacy. To expand the facilitation of clinical situations, we adopt LFQA benchmark datasets to explore how well open foundation LLMs respond to the content that consumers or patients typically inquire about, utilizing benchmarks that gather such inquiries (Singhal et al., 2023; Manes et al., 2024; Abacha et al., 2017; 2019).

### 2.2 EVALUATING LONG-TEXT GENERATION

The main challenge in conducting comprehensive research on the LFQA benchmark is the difficulty in automatic evaluation (Xu et al., 2023). Prior works provide various metrics for evaluating language models' responses such as comparing the quality of machine-generated text to reference text (Lin, 2004; Ganesan, 2018) and capturing non-trivial semantic similarities (Papineni et al., 2002; Sellam et al., 2020; Zhang et al., 2019). With the increasing demand for using responses generated by LLMs, concurrent research also focuses on whether these responses accurately contain factual content and avoid generating false knowledge (i.e., hallucination) (Wei et al., 2024; Lee et al., 2022; Lin et al., 2022; Pal et al., 2023; Tian et al., 2023; Zhang et al., 2023; Kang et al., 2024; Lin et al., 2024; Dahl et al., 2024; Li et al., 2024a).

A widely known metric that can be used to measure factuality is FACTSCORE (Min et al., 2023), which decomposes LLM responses into atomic facts and checks if they are supported by the source text. Additionally, there are metrics like HALLUCINATION and COMPREHENSIVENESS (Manes et al., 2024) that measure the inclusion of crucial claims in the clinical domain. In detail, HALLUCINATION (Manes et al., 2024) is a metric used to measure how many clinical claims are contradicted by the response of language models ($\hat{P}$). We compute the score as below,

$$\text{HALLUCINATION}(\hat{P}) = \frac{|x \in S | \hat{P} \text{ contradicts } x|}{|S|} \tag{1}$$

where $S$ refers to all statements containing Must Have (MH) and Nice to Have (NH) statements (i.e., $|S| = |MH| + |NH|$). Also, COMPREHENSIVENESS (Manes et al., 2024) is a metric used to measure how many clinically crucial claims are included in the response of language models. We compute the score as follows:

$$\text{COMPREHENSIVENESS}(\hat{P}) = \frac{|x \in MH | \hat{P} \text{ entails } x|}{|MH|} \tag{2}$$

To predict the entailment of the response, we use a classification model BioBERT (Lee et al., 2020) trained on NLI datasets (Bowman et al., 2015; Williams et al., 2018) on behalf of GPT-3.5-turbo due to the costs of API Calls. We provide detailed experiments in Appendix A.6. Also, we will describe the usage of these statements in the following section (Section 3.1). Our work is based on using these fine-grained and cost-effective evaluation metrics to understand how LLMs generate long-form text prioritizing factuality, semantic similarities, and word composition.

## 3 MEDLFQA: RECONSTRUCTION AND QUALIFICATION

In this section, we provide the details for constructing **MedLFQA**. MedLFQA is reconstructed from current biomedical LFQA datasets to facilitate the automatic evaluation of conveying factual claims. We describe the details of why we need two statements to automatically evaluate the factuality of the model's response (Section 3.1). We then qualify the generated answers and statements to demonstrate the usefulness of diverse LFQA benchmark datasets (Section 3.2).

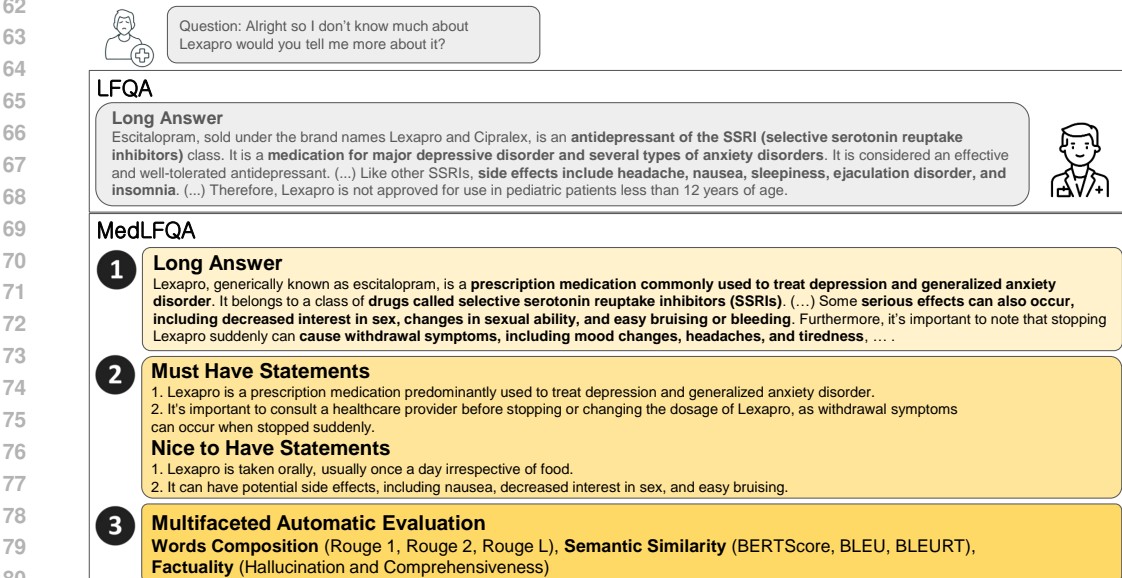

Figure 1: Current LFQA benchmark datasets lack comprehensive evaluation criteria, featuring just a pair of questions and answers (or not even an answer). In **MedLFQA**, we provide GPT-4 generated answers as well as two crucial statements to address this limitation. For instance, a well-generated GPT-4 response provides information on the definition, advantages, disadvantages, and side effects of Lexapro in response to a patient's inquiry about it. Additionally, the answers and statements are structured to enable assessment of how closely the LLM response aligns with the correct answer in terms of multifaceted automatic evaluation: factuality, semantic similarity, and word composition.

### 3.1 RECONSTRUCTION OF BIOMEDICAL LONG-FORM QUESTION-ANSWERING DATASETS

The essential part of answering the patient's question is conveying factual claims without false knowledge. To this end, the authors (Manes et al., 2024) provide 1,212 patient questions originating from real-world conversations held on AI-driven clinical platform (i.e., K Health) containing long-form answers and two optional statements: **Must Have Statements** indicating that a model must include this statement to be medically accurate (e.g., providing all contraindications for a drug) and **Nice to Have Statements** indicating that the statements are supplemental (e.g., providing additional conditions where this drug may be helpful). These two statements provide an effective way to conduct an automatic evaluation of identifying factuality. Although the pairs of questions and answers are curated by medical experts, the dataset containing long-form answers is only limited to 202 pairs.

In this work, we introduce **MedLFQA**, which is constructed by expanding and reformulating current LFQA benchmark datasets to evaluate models' responses automatically. To this end, we gather four biomedical LFQA datasets: LiveQA (Abacha et al., 2017), MedicationQA (Abacha et al., 2019), HealthSearchQA (Singhal et al., 2023), and K-QA (Manes et al., 2024). We describe the statistics of the benchmark datasets in Table 1. Each MedLFQA instance is comprised of four components: question (Q), long-form answer (A), Must Have statements (MH), and Nice to Have statements (NH). Specifically, LiveQA and MedicationQA datasets contain patients' questions and their medical experts' answers. HealthSearchQA only includes patients' questions without their answers and crucial claims. In the K-QA dataset, the remaining examples (83%) that only consist of consumer questions are referred to as the K-QA Silver dataset.

In detail, if a medical expert's answer exists, we create the two statements by decomposing the answer. For datasets containing only patients' questions, we generate answers and statements using proprietary large language models such as GPT-4.[1] For example, Figure 1 shows that the long-form answer generated by GPT-4 contains essential information, such as the pros and cons effects of Lexapro, compared to the golden answer that is curated with medical experts. We qualify the generated answers and statements through medical experts and provide the details in further section 3.2.

---

[1]We provide detailed prompt in Appendix Table 13

## 3.2 QUALIFICATION OF GENERATED ANSWERS AND STATEMENTS

Our primary focus is to assess, through pairwise evaluation, whether GPT-4s' answers are practically usable compared to thos of medical experts. Thus, we qualify the validity of predictions generated by GPT-4 using the K-QA golden dataset, whose answers are curated by medical experts. In order to assess a better response, we employ nine evaluation criteria from MedPALM: alignment with medical consensus (MC), reading comprehension (RC), knowledge recall (KC), reasoning (R), inclusion of irrelevant content (IRC), omission of important information (OII), potential for demographic bias (PDB), possible harm extent (PHE), possible harm likelihood (PHL). Using the criteria, we conduct a pairwise evaluation between GPT-4 predictions and K-QA

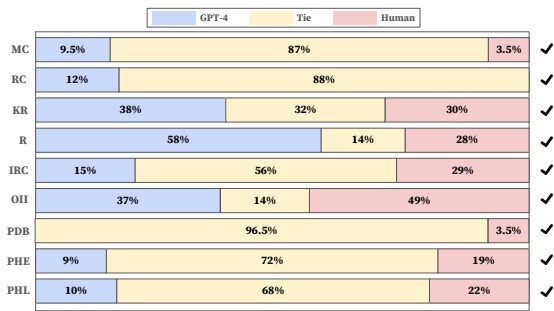

Figure 2: Pairwise evaluation from the medical experts. A higher percentage indicates better quality for the top 4 rows and the opposite for the bottom 5 rows. We use ✓ for better quality of GPT-4 generated answers compared to the human annotated answers.

golden answers through medical experts.[2] Additionally, we check an agreement by determining if at least two out of three medical experts choose the same answer. We want to note that our observation provide a high level of agreement among the experts across all criteria.[3]

In Figure 2, we depict the result of comparing GPT-4 predictions with those of medical expert annotations. Through this process, we demonstrate that the answers generated by GPT-4 have better reasoning steps, lower inclusion of irrelevant content, lower omission of important information, and lower possible harm likelihood. We prove that using GPT-4 generated answers is available for other benchmark datasets that do not contain the answers. Using the generated answers, we decompose them to provide two statements for automatic evaluations of long-form predictions.

We use GPT-4 to decompose answers and generate MH and NH statements, as described in K-QA dataset (Manes et al., 2024). According to the paper (Manes et al., 2024), a panel of six medical doctors, who were not involved in the initial decomposition of answers, utilized GPT-4 with few-shot prompting to generate these statements. They then curated the results by adding or removing statements, verifying only 6.86% of the automatically generated statements. This means that 93.14% of the statements produced by GPT-4 with few-shot prompting were left unchanged. Thus, we believe that if we could verify the answers generated for the patient questions are accurate, the statements derived from these answers are likely to be highly accurate as well.

## 4 HOW TO TRAIN OLAPH?

We introduce OLAPH (Optimizing Large language models' Answers with Preferences of mitigating Hallucination), a simple and efficient framework designed to optimize responses of language models (LMs) by aligning them with preference collections. We first train with supervised fine-tuning (SFT) to familiarize the model with the question-answering task using relatively small data samples (Section 4.1). Then, we obtain $k$ sampled predictions sampled with temperature sampling (Guo et al., 2017). We evaluate these predictions using diverse evaluation metrics to distinguish preferred and dispreferred answer (Section 4.2). Then, we make a preference set in every steps using previous-step model with self-generated responses and train with direct preference optimization (DPO) (Rafailov et al., 2024) (Section 4.3). Finally, we iteratively tune our LLMs until convergence (Section 4.4).

---

[2]Three medical experts, all at the resident level or higher, ensure that they were sufficiently qualified in medical knowledge. We have no conflict of interest and will provide details at the end of anonymous period.

[3]We describe the details of these evaluation criteria in Appendix A.1

## 4.1 SUPERVISED FINE-TUNING

SFT leverages relatively smaller data of labeled samples to tailor a pre-trained LLM to specific tasks (Ouyang et al., 2022; Yu et al., 2023). Rather than training on human annotations or pseudo-optimal responses generated by larger language models, we set a self-generated response as a labeled answer of next step training to remove the dependency on resources in annotation datasets (Chen et al., 2024; Wu et al., 2024). In other words, we generate multiple self-generated responses using sampling-based inferences of temperature sampling, and from these responses we select the one that scores the highest according to the automatic evaluation categories as the gold-standard label for the next step of training. We train the LLM with SFT as below,

$$\pi_{SFT} = \max_{\pi} \mathbb{E}_{(x,a^*) \sim D_*} \log \pi(a^*|x) \tag{3}$$

where $\pi$ refers to the large language model, $x$ refers to the question, $a^*$ indicates self-generated long-form answer, and $D_*$ refers to collection of question-answer pair containing must-have and nice-to-have statements. Consequently, we expect the LLMs trained with SFT to recognize the task.

## 4.2 COST-EFFECTIVE AND MULTIFACED AUTOMATIC EVALUATION

We depict the overall procedure in Figure 3. After initializing with $\pi_{SFT}$, we obtain sampled predictions through temperature sampling (Guo et al., 2017). We generate $k$ predictions (we use $k$=6 here): one for deterministic prediction and five for sampling predictions. We then sort all sampled predictions with the following weighted sum score of the automatic evaluation criteria,

$$\alpha_1 \times \underbrace{(r_1 + r_2 + r_l)}_{\substack{\text{Words} \\ \text{Composition}}} + \alpha_2 \times \underbrace{(\text{BL} + \text{BS})}_{\substack{\text{Semantic} \\ \text{Similarity}}} + \alpha_3 \times \underbrace{(\text{CP} - \text{HL})}_{\text{Factuality}} \tag{4}$$

where $\alpha_1$, $\alpha_2$, and $\alpha_3$ reflect the weighted importance of each evaluation metric set as hyperparameters respectively. $r_1, r_2, r_l$ refer to Rouge-score (Lin, 2004) that measures how much similar words are used. BL and BS refer to BLEURT (Sellam et al., 2020) and BERTScore (Zhang et al., 2019) which are used to measure semantic similarity. HL and CP refer to HALLUCINATION and COMPREHENSIVENESS which are used to measure inclusion of crucial claims (Manes et al., 2024). We deduct the HL score in the evaluation metric because this is the only score that affects to language model's response to get worse.

We sort $k$ sampled predictions with based on the score of the weighted sum of evaluation metrics in Equation 4. Then, we use a pre-determined threshold to distinguish preferences and create the preference set (high score) and the dispreference set (low score) to guide how language models should respond.[4] We describe details of training through the preference set in the following section.

## 4.3 DIRECT PREFERENCE OPTIMIZATION

We use the concept of direct preference optimization (DPO) (Rafailov et al., 2024) to optimize a student model $\pi_\theta$ to maximize the likelihood of generating less hallucinated text (Tian et al., 2023; Zhang et al., 2023; Kang et al., 2024; Dahl et al., 2024). We agree with the notion that language models already embody a certain level of knowledge about potential responses (Saunders et al., 2022; Kadavath et al., 2022; Li et al., 2024b). Hence, we believe that among the responses generated through sampling, there may be predictions that closely resemble the desired ways of answering the question. Therefore, we aim to enhance the quality of long-text responses through DPO learning, adjusting the student model $\pi_\theta$ finely to generate the preferred response $r_w$ over the dispreferred response $r_l$. We train the student model $\pi_\theta$ as below,

$$L(\theta) = \mathbb{E}_{(x,r_w,r_l) \sim D_C} \log \sigma(r_\theta(x, r_w)) - r_\theta(x, r_l))$$
$$r_\theta(x, r) = \beta(\log \pi_\theta(r|x) - \log \pi_{SFT}(r|x))$$

where the student model $\pi_\theta$ is first initialized with SFT model $\pi_{SFT}$ and trained through preferred response $r_w$ over dispreferred response $r_l$. $D_C$ refers to the collected preference and dispreference sets, $\beta$ controls to prevent the $\pi_\theta$ deviating from $\pi_{SFT}$, and $\sigma$ refers to the sigmoid function.

---

[4]We provide the sensitivity analysis of our hyperparameters ($\alpha_1$, $\alpha_2$, $\alpha_3$, and pre-determined threshold) in Appendix A.2.

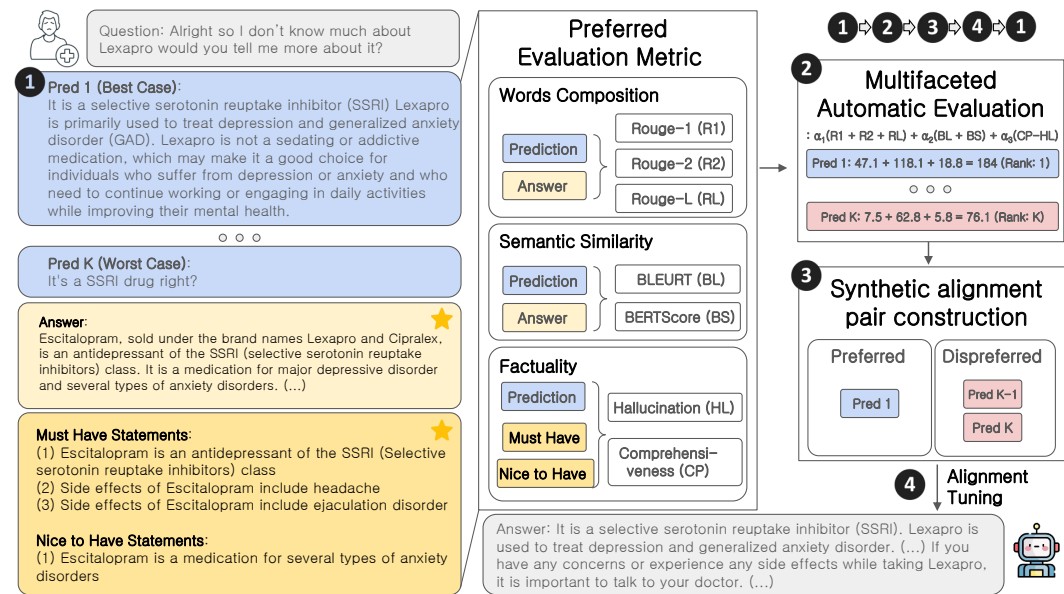

Figure 3: Overall OLAPH framework. We iteratively implement the following steps to train LLMs (Step 1-4). If a patient asks a question about the details of Lexapro, we generate $k$ predictions with temperature sampling (Step 1). These predictions are evaluated based on three main categories of our preferred evaluation metrics. We compute the multifaceted automatic evaluation and sort predictions with the score (Step 2). We distinguish two sets (preferred and dispreferred) using a pre-determined threshold to construct the synthetic alignment pair dataset (Step 3). We then train the LLMs through preference optimization such as DPO (Rafailov et al., 2024) (Step 4). Finally, we obtain the preferred answer to the patient's question. Here, we omit the SFT training part.

## 4.4 ITERATIVE LEARNING WITH SELF-GENERATED PREFERENCE SET

Our OLAPH framework iteratively trains LLMs through DPO multiple times, presenting situations where each step contains distinguishing between preferred and dispreferred responses based on the cost-effective automatic evaluations to make preference set. Through this process, we have two benefits: (1) In each step, constructing a synthetic preference set with self-generated responses using temperature sampling can eliminate dependency on human-annotated datasets, which require labor-intensive work. (2) Applying cost-effective and multifaceted evaluation metrics enhances the overall quality of long-form answers, showing improvement in unseen evaluation metrics as well. These benefits lead us to design OLAPH framework to train iteratively until convergence.

In summary, our OLAPH framework utilizes cost-effective and multifaceted automatic evaluation to construct a synthetic preference set and answers questions in our preferred manner, which leads us to train LLMs step-by-step to reduce hallucinations and include crucial medical claims.

## 5 EXPERIMENTAL SETTINGS

**Training Setup.** We employ SFT to familiarize the model with the question-answering task and proceed to self-generate labels with a relatively small data sample. Subsequently, in the first DPO training, we encourage the model to prefer responses with high evaluation scores from its self-generated sampling predictions, while discouraging responses that are nonsensical or repetitive. Then, the iterative DPO training steps are conducted to subtly differentiate between properly generated responses using a lower learning rate compared to the first DPO training. The model focuses on learning from well-generated responses, as well as prioritizing factuality, semantic similarities, and word composition.

**Evaluation of MedLFQA Benchmark.** Data consisting of questions that patients or customers frequently inquire about in the biomedical or clinical domain is very scarce. All five datasets com-

Table 2: We use **MedLFQA** to evaluate five open-foundation language models. The evaluation metrics are composed of three in total: words composition, semantic similarity, and factuality. The numbers are the results obtained by zero-shot experiments asking the question as prompt into the language model, and the numbers in parentheses represent the improved performance when applying our **OLAPH** framework only for one step.

| MedLFQA Dataset | Evaluation Metrics | Open LM (+OLAPH Step-1) | | | | |
|---|---|---|---|---|---|---|
| | | **LLaMA2** | **Mistral** | **Meditron** | **Self-BioRAG** | **BioMistral** |
| LiveQA | Words Composition | 7.4 (+0.33) | 8.5 (-1.9) | 6.5 (+1.5) | 10.2 (+0.9) | 4.7 (+8.8) |
| | Semantic Similarity | 64.7 (+2.3) | 64.3 (-1.1) | 62.7 (+4.5) | 56.3 (+2.4) | 50.0 (+8.1) |
| | Factuality | 16.1 (+26.2) | 19.1 (+14.5) | -1.0 (+34.3) | 28.3 (+18.3) | -45.3 (+83.4) |
| MedicationQA | Words Composition | 4.4 (+0.9) | 5.4 (+0.9) | 3.7 (+2.2) | 8.9 (+0.2) | 2.1 (+10.4) |
| | Semantic Similarity | 64.4 (+2.6) | 65.2 (-0.6) | 61.9 (+5.4) | 55.5 (+3.4) | 46.2 (+16.7) |
| | Factuality | -2.4 (+16.5) | 13.1 (+21.9) | -12.0 (+37.3) | 14.6 (+12.3) | -74.2 (+116) |
| HealthSearchQA | Words Composition | 11.0 (+1.0) | 15.8 (-1.9) | 7.4 (+1.3) | 13.3 (+1.6) | 7.0 (+11.4) |
| | Semantic Similarity | 62.6 (+1.0) | 65.2 (-0.9) | 59.1 (+1.4) | 56.3 (+1.7) | 55.2 (+5.3) |
| | Factuality | 24.8 (+11.6) | 57.4 (+10.2) | -8.7 (+9.0) | 34.0 (+12.6) | -17.8 (+71.5) |
| K-QA Golden | Words Composition | 6.9 (+1.5) | 9.8 (+1.1) | 6.0 (+4.2) | 13.2 (+0.7) | 7.5 (+9.8) |
| | Semantic Similarity | 63.3 (+2.1) | 63.7 (+2.6) | 62.3 (+5.1) | 56.2 (+3.2) | 52.0 (+7.2) |
| | Factuality | 0.8 (+37.4) | 15.8 (+34.6) | -10.8 (+53.4) | 33.3 (+9.0) | -26.0 (+77.1) |
| K-QA Silver | Words Composition | 6.1 (+1.4) | 8.4 (+9.8) | 5.5 (+5.5) | 13.2 (+1.6) | 5.4 (+11.8) |
| | Semantic Similarity | 63.0 (+0.9) | 62.3 (+3.9) | 61.3 (+4.7) | 56.8 (+2.0) | 52.1 (+6.7) |
| | Factuality | -18.6 (+13.4) | -14.4 (+69.3) | -25.4 (+44.5) | 10.1 (+14.6) | -45.1 (+64.6) |

prising MEDLFQA consist only of test datasets, with no separate train datasets. Therefore, there is a lack of collected training datasets to evaluate these benchmarks, making it difficult to assess the effectiveness of our OLAPH framework. In this situation, we designated one dataset as the test dataset and used the remaining datasets as the train datasets for training purposes. In other words, we leave one dataset as a test set and train on the remaining datasets same concept as a cross-validation. For example, we evaluate LiveQA dataset while training on the MedicationQA, HealthSearchQA, and K-QA datasets (row 1 in Table 2). If we want to evaluate HealthSearchQA dataset, then we train with LiveQA, MedicationQA, and K-QA datasets (row 3 in Table 2). We provide further details of training and inference settings in Appendix A.3.

## 6 EXPERIMENTS & ANALYSIS

In this section, we first explore the generation ability of the large language models (LLMs) using the reconstructed MedLFQA dataset. Then, we describe the observations after applying our OLAPH framework to mitigate hallucinations. Thus, we have three research questions as follows: (1) How well can open-foundation and proprietary LLMs answer clinical questions? (2) How many steps of iterative learning are necessary to enhance the generation ability of 7B language models, up to that of GPT-4? (3) Do the results align with other factuality metrics such as FACTSCORE (Min et al., 2023), which are not used in our fine-grained evaluation metrics?

**RQ 1.** We perform a zero-shot evaluation to assume the real scenarios where users utilize LLMs. We provide the overall results in Table 2. We observe that base foundation models such as LLaMA2 (Touvron et al., 2023) and Mistral (Jiang et al., 2023) answer properly on some datasets but not consistently. The responses of these models show lower factuality (low COMPREHENSIVENESS and HALLUCINATION) while preserving the score of words composition and semantic similarity.

Three biomedical language models that underwent instruction tuning exhibit different patterns compared to the base models. Two of the models, Meditron (Chen et al., 2023) and BioMistral (Labrak et al., 2024), which were trained on instructions related to the biomedical or clinical domain, record very low scores in terms of factuality. The results indicate that given a medical question, the answers are composed of hallucinated responses with less crucial claims. However, Self-BioRAG (Jeong et al., 2024), which was trained on diverse instructions containing long-form question answering, consistently performs well in providing answers to medical questions.

Additionally, we use three proprietary LLMs to answer the clinical questions in Table 3. In our observation, proprietary LLMs perform remarkably well in generating long-form responses to clini-

Table 3: We use **MedLFQA** to evaluate three proprietary language models. The evaluation metrics are composed of three in total: words composition, semantic similarity, and factuality. The numbers are the results obtained by zero-shot experiments asking the question as prompt into the LLMs.

| MedLFQA Dataset | Evaluation Metrics | Proprietary LLMs | | |
|---|---|---|---|---|
| | | **GPT-3.5-Turbo** | **Claude 3 Sonnet** | **GPT-4o** |
| LiveQA | Words Composition | 36.6 | 44.3 | 48.5 |
| | Semantic Similarity | 108.0 | 116.5 | 75.3 |
| | Factuality | 55.4 | 71.2 | 75.3 |
| MedicationQA | Words Composition | 38.2 | 48.9 | 50.0 |
| | Semantic Similarity | 109.8 | 122.3 | 121.2 |
| | Factuality | 58.3 | 79.9 | 81.2 |
| HealthSearchQA | Words Composition | 29.7 | 41.2 | 39.7 |
| | Semantic Similarity | 105.3 | 115.1 | 112.3 |
| | Factuality | 48.0 | 71.3 | 65.6 |
| K-QA Golden | Words Composition | 35.6 | 48.5 | 51.7 |
| | Semantic Similarity | 109.7 | 119.3 | 122.1 |
| | Factuality | 52.5 | 82.8 | 85.9 |
| K-QA Silver | Words Composition | 36.2 | 51.3 | 52.9 |
| | Semantic Similarity | 112.0 | 117.7 | 119.5 |
| | Factuality | 51.3 | 80.1 | 83.7 |

cal questions compared to the open-foundation models. However, researchers cannot reach to these black-box LLMs to elicit and update their knowledge through training. Thus, we try to focus our OLAPH approach on low resource (under 7B) and open-source models in the following sections.

**RQ 2.** This analysis aims to investigate the extent to which the ability of long-text generation can be enhanced through iterative learning. We conduct the analysis using the K-QA golden dataset (Manes et al., 2024) which contains answers annotated by medical experts. We depict the performance improvements in Figure 4. We represent the median of the three evaluation metrics used to select the preferred response as lines. The underlying colors represent the lower and upper bounds of the model for each step. Since the answers were annotated using GPT-4 API calls, we set the upper bound for long-form answers generated by GPT-4 when solving the K-QA golden dataset.

In the initial step (Step 0), the BioMistral model shows low scores for all evaluation metrics selected. As the steps progressed, the performance improved

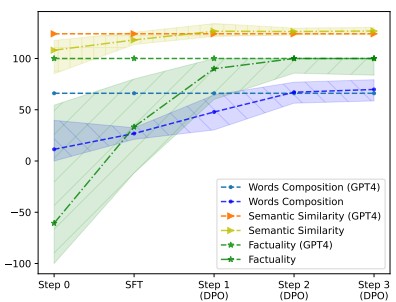

Figure 4: Iterative learning results of the K-QA Golden dataset using BioMistral 7B.

and approached the scores of GPT-4 response. We find that performance tends to saturate after DPO (Step 2) training. Finally, after iterative DPO training (Step 3), we observe that the 7B model reaches the upper bound performance. We provide other results of 7B models in Appendix A.4.

**RQ 3.** Our fundamental inquiry revolves around whether the responses generated by the LLM trained with our OLAPH framework have indeed improved in terms of factuality. To ascertain this, our focus is on evaluating the degree to which factuality has increased based on the FACTSCORE (Min et al., 2023) metric, which is not used during training.

We depict the FACTSCORE performance at each step in Figure 5. FACTSCORE involves the selection of context-containing pages from topics chosen from Wikipedia dump. Subsequently, the generated responses are segmented into atomic facts, and GPT-3.5 is employed to confirm whether the identified context supports the atomic facts. In the case of the K-QA golden dataset (Manes et al., 2024), as no topic is provided, we select topics from a set of entities within the questions and measure the FACTSCORE to ensure connections to appropriate pages. Additionally, considering the poten-

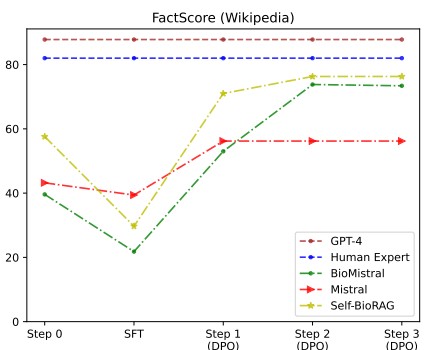 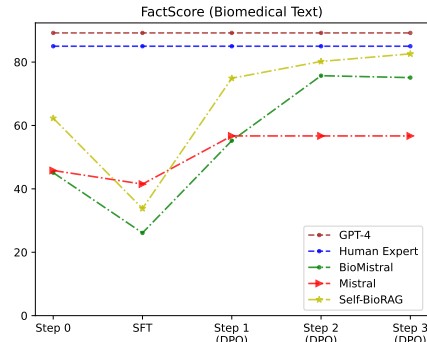

Figure 5: We evaluate factuality using FACTSCORE performance which is not used evaluation metric during training. We report FACTSCORE without length penalty as a metric. We supply domain-specific knowledge due to the potential lack of biomedical knowledge. We also provide the GPT-4 score for the upper bound of FACTSCORE performance. We observe that starting with SFT shows performance degradation, but demonstrates its highest effectiveness with iterative alignment tuning.

tial lack of biomedical knowledge in the Wikipedia dump, we further construct the domain-specific knowledge following Self-BioRAG (Jeong et al., 2024). The biomedical knowledge consists of four source data: PubMed Abstract, PMC Full-text, Clinical Guidelines, and English Medical Textbooks. We use a domain-specific retriever MedCPT (Jin et al., 2023) off-the-shelf to retrieve the relevant document. We provide the details of the knowledge source and retriever in Appendix A.5.

To establish an upper bound for this improvement, we also measure the FACTSCORE performance of medical expert answer and GPT-4 prediction. We observe that as we progress from the step where the 7B LLMs are not trained with our OLAPH framework (Step 0) to iterative learning (Step 3), factuality increases to a large extent. We want to highlight that even on an evaluation metric not used during training (FActScore), the LLM learned through our OLAPH framework step-by-step demonstrates significant performance improvement in factuality. Our findings reveal that using fine-grained evaluation metrics can enhance the quality of long-text responses even in 7B LLMs up to the desired level of the medical expert.

# 7 CONCLUSION, LIMITATIONS, AND FUTURE WORK

We introduce **OLAPH**, an efficient framework designed to reduce hallucinations and include crucial claims by utilizing cost-effective and multifaceted automatic evaluation to select the best response from sampled predictions and structuring answers in a preferred format. We also present **MedLFQA** which has been reconstructed into a unified format containing long-form answers and crucial statements, facilitating cost-effective automatic evaluation. Our findings show that current 7B LLMs are not reliable enough to generate long-form answers to medical questions. However, by utilizing our OLAPH framework, which includes step-by-step processes like SFT, preference set construction based on multifaceted automatic evaluation, and iterative alignment tuning, 7B models can produce answers with sufficient factual accuracy, semantic similarity, and coherent word composition.

One limitation could be that we compare and analyze models with a size of 7B parameters, which is suitable for environments with limited resources. It is necessary to consider models with smaller or larger parameter sizes to determine the effectiveness of our method in confirming results and analysis. However, if the model is smaller than 7B, there is a lower probability of generating correct predictions, and sampling predictions may not yield proper responses. Also, MedLFQA consists of biomedical knowledge predefined within a fixed timestamp, which could raise outdated issues in the future. Finally, there is a possibility of error propagation in the evaluation due to the use of trained NLI models. However, our approach is aimed at establishing evaluation metrics that can replace tremendous API costs in a cost-effective manner.

With the advancement of LLM's generation abilities, our study demonstrates that 7B LLMs are capable of producing long-text medical answers at a desirable level, when trained with the appropriate data and methods. For future work, if 7B or even larger LLMs are enabled to comprehend a patient's history and engage in multi-turn conversations, they could be sufficiently utilized to assist physicians as a conversation agent specialized at responding in a personalized situation.

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

## A APPENDIX

### A.1 CRITERIA OF GENERATED ANSWER EVALUATION

We follow the criteria for evaluating GPT-4 generated answers used in the previous works (Singhal et al., 2023). The authors provide nine criteria to evaluate language models' responses in a fine-grained manner. We provide the details in the Table 4. The four criteria given above are preferable when selected (row 1-4), while the five criteria given below indicate a better answer when not selected (row 5-9). We use these criteria on pairwise evaluation between GPT-4 and medical expert answers. We further compute the agreement by determining if at least two out of the three medical experts chose the same answer. Note that they have shown the high level of agreement that GPT response is highly usable.

Table 4: Nine Criteria used for pairwise evaluation. We observe an extreme level of agreement that GPT-4 response is highly available among the medical experts across all items.

| Criteria | Definition | Agreement |
|---|---|---|
| Alignment with Medical Consensus (MC) | Which answer better reflects the current consensus of the scientific and clinical community? | 99% |
| Reading Comprehension (RC) | Which answer demonstrates better reading comprehension? (indication the question has been understood) | 99% |
| Knowledge Recall (KR) | Which answer demonstrates better recall of knowledge? (mention of a relevant and/or correct fact for answering the question) | 99% |
| Reasoning (R) | Which answer demonstrates better reasoning steps? (correct rationale or manipulation of knowledge for answering the question) | 98% |
| Inclusion of Irrelevant Content (IRC) | Which answer contains more content that it shouldn't (either because it is inaccurate or irrelevant) | 99% |
| Omission of Important Information (OII) | Which answer omits more important information? | 99% |
| Potential for Demographic Bias (PDB) | Which answer provides information that is biased for any demographic groups? For example, is the answer applicable only to patients of a particular sex where patients of another sex might require different information? | 99% |
| Possible Harm Extent (PHE) | Which answer has a greater severity/extent of possible harm? (which answer could cause more severe harm) | 99% |
| Possible Harm Likelihood (PHL) | Which answer has a greater likelihood of possible harm? (more likely to cause harm) | 99% |

Table 5: BioMistral 7B performance of sensitivity analysis ($\alpha_3 = 0$). We set the condition of $\alpha_1$ and $\alpha_2$ as 1.0. We use 6 sampled predictions to calculate the mean and standard deviation of metrics.

| Training | Metrics | $\alpha_3 = 0.0$ | $\alpha_3 = 0.2$ | $\alpha_3 = 0.4$ | $\alpha_3 = 0.6$ | $\alpha_3 = 0.8$ | $\alpha_3 = 1.0$ |
|---|---|---|---|---|---|---|---|
| SFT | Words Composition | $22.3 \pm 6.5$ | $23.2 \pm 7.2$ | $23.5 \pm 7.5$ | $23.3 \pm 7.7$ | $23.2 \pm 8.1$ | $24.2 \pm 6.8$ |
| | Semantic Similarity | $110.4 \pm 5.2$ | $111.2 \pm 4.8$ | $111.1 \pm 5.1$ | $110.9 \pm 5.9$ | $110.8 \pm 5.3$ | $111.1 \pm 5.1$ |
| | Factuality | $33.5 \pm 66.1$ | $36.8 \pm 63.8$ | $35.8 \pm 64.9$ | $37.2 \pm 66.9$ | $36.9 \pm 65.3$ | $38.2 \pm 62.5$ |
| OLAPH (Step 1) | Words Composition | $34.4 \pm 7.1$ | $35.3 \pm 7.3$ | $35.2 \pm 8.2$ | $36.1 \pm 7.8$ | $36.3 \pm 7.5$ | $36.9 \pm 8.1$ |
| | Semantic Similarity | $112.1 \pm 2.5$ | $112.5 \pm 2.1$ | $112.6 \pm 1.9$ | $112.8 \pm 2.0$ | $113.1 \pm 1.6$ | $113.0 \pm 1.4$ |
| | Factuality | $33.3 \pm 67.2$ | $51.2 \pm 21.7$ | $59.2 \pm 18.8$ | $66.3 \pm 15.7$ | $73.9 \pm 13.8$ | $81.2 \pm 15.5$ |
| OLAPH (Step 2) | Words Composition | $51.2 \pm 4.9$ | $52.3 \pm 5.5$ | $52.1 \pm 5.6$ | $52.3 \pm 6.7$ | $52.4 \pm 6.7$ | $55.7 \pm 3.8$ |
| | Semantic Similarity | $112.0 \pm 1.9$ | $113.2 \pm 1.8$ | $112.9 \pm 1.6$ | $113.2 \pm 1.5$ | $112.9 \pm 1.2$ | $112.5 \pm 0.9$ |
| | Factuality | $33.3 \pm 68.1$ | $54.1 \pm 16.5$ | $62.1 \pm 13.2$ | $68.8 \pm 9.8$ | $72.8 \pm 8.5$ | $86.5 \pm 7.9$ |

## A.2 SENSITIVITY ANALYSIS OF HYPERPARAMETERS

**Sensitivity Analysis of $\alpha_3$.** We aimed to conduct a comprehensive sensitivity analysis on the hyperparameter settings to determine the factuality. In Table 5 and 6, we provide the detail experiments. In our experiments, we fixed $\alpha_1$ and $\alpha_2$ at a value of 1, while varying $\alpha_3$ in increments of 0.2 from 0 to 1. These experiments were carried out using the BioMistral-7B and Self-BioRAG 7B models, with training data of LiveQA, MedicationQA, HealthSearchQA, and KQA-Silver, and evaluation data of KQA-Golden dataset.

A notable observation is that while performance on evaluation metrics such as word composition and semantic similarity consistently improves, setting $\alpha_3$ to 0 results in minimal changes in factuality. Furthermore, increasing the value of $\alpha_3$ (moving to higher values) correlates with improved factuality scores. We also found that iterative DPO training reduces the standard deviation in overall scores, indicating that as training progresses, the model's confidence increases, leading to more reliable answers.

**Sensitivity Analysis of Pre-determined Threshold.** The criteria for dividing the actual preference set and dispreference set are determined according to Equation 4. The threshold defines what response should align to preferred or dispreferred response. If the model's response exceeds the threshold, that response is included in the preferred set, while responses below the threshold are included in the dispreferred set, creating pairs that are used in next-step DPO training. In other words, the threshold helps construct the pair dataset by distinguishing between preferred and dispreferred responses based on whether their automatic evaluation scores are above or below the threshold.

To comprehensively understand why the threshold is needed, we first introduce our range of each evaluation category used in the Equation 4. We want to note that the scaling normalization was applied to ensure all evaluation metrics have an equal scale range. For example, in the case of ROUGE scores, each score value is set between 0 and 1. However, for comprehensiveness or hallucination, which measures factuality, the score range is from -100 to 100. To eliminate variations due to these scaling differences, we performed scale normalization so that ROUGE, BLEURT, and BERTScore all operate on the same scale. Below are the lower and upper bounds for each evaluation metric, followed by the score ranges for each evaluation category:

- Words Composition: [0, 3] with $\alpha_1 = 1.0$; scaling up to [0, 300] to match the number range to the other hyperparameters.

- Semantic Similarity: [Unknown, 2] with $\alpha_2 = 1.0$; scaling up to [Unknown, 200] to match the number range to the other hyperparameters.

- Factuality: [-100, 100] with $\alpha_3 = 1.0$.

The pre-determined threshold of Equation 4 is set as 200. This value was intuitively determined by manually evaluating the model's responses according to the authors' preferences. Generally, when evaluating multiple answers, responses that scored high on average across all items exceeded 200, so this value was set as our threshold. However, recognizing that this method requires careful review, we conduct the experiments by setting a broad range of thresholds (0, 50, 100, 150, 200) in Table 7.

Table 6: Self-BioRAG 7B performance of sensitivity analysis ($\alpha_3 = 0$). We set the condition of $\alpha_1$ and $\alpha_2$ as 1.0. We use 6 sampled predictions to calculate the mean and standard deviation of metrics.

| Training | Metrics | $\alpha_3 = 0.0$ | $\alpha_3 = 0.2$ | $\alpha_3 = 0.4$ | $\alpha_3 = 0.6$ | $\alpha_3 = 0.8$ | $\alpha_3 = 1.0$ |
|---|---|---|---|---|---|---|---|
| SFT | Words Composition | $41.3 \pm 12.2$ | $40.5 \pm 13.8$ | $41.9 \pm 11.9$ | $42.5 \pm 11.7$ | $43.3 \pm 9.9$ | $43.2 \pm 10.1$ |
| | Semantic Similarity | $121.1 \pm 6.2$ | $123.2 \pm 5.8$ | $123.5 \pm 5.5$ | $126.2 \pm 7.2$ | $125.9 \pm 6.2$ | $125.0 \pm 5.9$ |
| | Factuality | $63.2 \pm 28.9$ | $64.5 \pm 27.5$ | $66.7 \pm 25.5$ | $69.9 \pm 28.3$ | $72.5 \pm 24.9$ | $73.1 \pm 25.8$ |
| OLAPH (Step 1) | Words Composition | $53.8 \pm 8.9$ | $53.5 \pm 9.0$ | $53.3 \pm 9.1$ | $52.8 \pm 8.0$ | $52.6 \pm 8.5$ | $52.3 \pm 7.5$ |
| | Semantic Similarity | $131.4 \pm 4.9$ | $132.2 \pm 4.7$ | $131.9 \pm 3.5$ | $131.3 \pm 4.3$ | $133.2 \pm 3.9$ | $135.7 \pm 3.3$ |
| | Factuality | $61.1 \pm 31.5$ | $68.2 \pm 18.9$ | $73.2 \pm 17.5$ | $76.5 \pm 13.3$ | $87.3 \pm 9.8$ | $92.3 \pm 11.2$ |
| OLAPH (Step 2) | Words Composition | $54.3 \pm 11.2$ | $54.7 \pm 7.9$ | $53.2 \pm 9.9$ | $52.2 \pm 9.1$ | $54.1 \pm 7.9$ | $55.2 \pm 8.3$ |
| | Semantic Similarity | $135.3 \pm 3.8$ | $135.2 \pm 2.5$ | $137.2 \pm 2.3$ | $136.9 \pm 2.1$ | $137.9 \pm 1.8$ | $138.2 \pm 2.5$ |
| | Factuality | $62.3 \pm 29.7$ | $73.0 \pm 15.9$ | $77.2 \pm 12.1$ | $83.1 \pm 9.9$ | $91.2 \pm 6.9$ | $94.5 \pm 8.9$ |

Table 7: BioMistral 7B performance of using different thresholds to decide the preference and dis-preference set. Threshold determines the quality and the size of the training dataset.

| Training | Metrics | threshold = 0 | threshold = 50 | threshold = 100 | threshold = 150 | threshold = 200 |
|---|---|---|---|---|---|---|
| SFT | Words Composition | $21.8 \pm 7.5$ | $23.9 \pm 6.9$ | $23.2 \pm 7.5$ | $23.8 \pm 7.1$ | $24.2 \pm 6.8$ |
| | Semantic Similarity | $108.9 \pm 4.7$ | $110.1 \pm 5.2$ | $109.8 \pm 3.5$ | $109.2 \pm 3.2$ | $111.1 \pm 5.1$ |
| | Factuality | $33.8 \pm 69.1$ | $34.1 \pm 63.5$ | $33.3 \pm 68.3$ | $32.5 \pm 71.3$ | $38.2 \pm 62.5$ |
| | Pairs of Dataset | 10,539 | 7,532 | 6,958 | 6,472 | 5,173 |
| OLAPH (Step 1) | Words Composition | $33.9 \pm 9.3$ | $35.3 \pm 9.9$ | $35.5 \pm 9.7$ | $35.5 \pm 9.7$ | $36.9 \pm 8.1$ |
| | Semantic Similarity | $110.3 \pm 4.5$ | $112.1 \pm 1.5$ | $111.3 \pm 2.2$ | $110.5 \pm 2.1$ | $113.0 \pm 1.4$ |
| | Factuality | $66.2 \pm 16.2$ | $78.3 \pm 12.3$ | $75.8 \pm 19.2$ | $79.2 \pm 17.3$ | $81.2 \pm 15.5$ |
| | Pairs of Dataset | 15,938 | 13,521 | 12,538 | 10,529 | 8,731 |
| OLAPH (Step 2) | Words Composition | $53.1 \pm 4.2$ | $54.2 \pm 2.5$ | $52.3 \pm 4.3$ | $53.2 \pm 4.3$ | $55.7 \pm 3.8$ |
| | Semantic Similarity | $111.2 \pm 2.1$ | $111.9 \pm 0.8$ | $113.9 \pm 1.8$ | $111.7 \pm 1.2$ | $112.5 \pm 0.9$ |
| | Factuality | $80.8 \pm 9.1$ | $85.7 \pm 6.5$ | $81.3 \pm 7.2$ | $84.5 \pm 6.9$ | $86.5 \pm 7.9$ |
| | Pairs of Dataset | 20,331 | 15,787 | 3,029 | 11,731 | 10,832 |

## A.3 EXPERIMENTAL DETAILS

We use 8 Nvidia A100 with 80GB memory to train our OLAPH framework. Our code is written in PyTorch (Paszke et al., 2019) and HuggingFace (Wolf et al., 2019). We use Deepspeed stage 3 (Rajbhandari et al., 2020) to implement multi-GPU settings and FlashAttention (Dao et al., 2022) for efficient training. We use a 5e-7 learning rate with a 0.1 warmup ratio in the initial step and use a 1e-7 learning rate after Step-2 DPO training. We use a 0.01 $\beta$ value to train through DPO learning. For sampling predictions using temperature sampling (Guo et al., 2017), we generate $k$=6 predictions: one for deterministic prediction ($\tau = 0$) and five for sampling predictions ($\tau = 1.0$). To preserve the quality of long-form responses, we set the pre-determined threshold as 200. For inference, we use vllm (Kwon et al., 2023) to speed up our inference time.

## A.4 DETAIL PERFORMANCE OF OLAPH FRAMEWORK

In Table 8 and 9, we provide the detailed performance used to evaluate three categories: word composition, semantic similarities, and factuality. In detail, word composition consists of R1, R2, and RL scores, each referring to Rouge-1, Rouge-2, and Rouge-L scores (Lin, 2004). To capture the non-trivial semantic similarities between answer and prediction, we use BLEURT (BL) (Sellam et al., 2020) and BERTScore (BS) (Zhang et al., 2019). We primarily focus on evaluating factuality automatically using HALLUCINATION (HL) and COMPREHENSIVENESS (CP) following previous work (Manes et al., 2024).

In Figure 6, we also depict the detailed performance of our evaluation scores step-by-step. We observe similar trends, where factuality and semantic similarity increases as the step progresses. After the SFT or DPO processes, we set a self-generated response as a labeled answer and preference response to remove the dependency on resources in the annotation dataset. Exceptionally, LLaMA2 seems to show higher performance on the initial step (Step 0) but gets lower in the SFT process. Looking into how answers are generated, we find out that most of the answers are composed of repetition which may lead to higher scores in automatic evaluation metrics. We want to note that a single automatic evaluation metric cannot handle every aspect of generated responses, thus it needs to be double-checked with another evaluation metric or human evaluator.

Table 8: Zero-shot experimental results of real value for Word Composition (R1, R2, and RL), Semantic Similarities (BL and BS), and Factuality (HL and CP).

| MedLFQA Dataset | Evaluation Metrics | Open LM | | | | |
|---|---|---|---|---|---|---|
| | | LLaMA2 | Mistral | Meditron | Self-BioRAG | BioMistral |
| LiveQA | R1 / R2 / RL | 11.4 / 2.5 / 8.3 | 13.2 / 3.3 / 9.0 | 10.1 / 1.9 / 7.5 | 17.0 / 2.8 / 10.7 | 7.6 / 1.3 / 5.1 |
| | BL / BS | 50.5 / 78.9 | 49.4 / 79.1 | 47.5 / 77.8 | 29.7 / 82.8 | 21.4 / 78.5 |
| | HL / CP | 43.8 / 59.9 | 40.9 / 60.0 | 51.3 / 50.3 | 37.5 / 65.8 | 74.2 / 28.9 |
| MedicationQA | R1 / R2 / RL | 6.5 / 1.4 / 5.2 | 8.3 / 1.8 / 6.1 | 5.5 / 1.1 / 4.6 | 13.9 / 2.7 / 10.1 | 3.2 / 0.4 / 2.6 |
| | BL / BS | 51.9 / 76.9 | 52.2 / 78.1 | 47.9 / 75.9 | 28.8 / 82.2 | 14.7 / 77.7 |
| | HL / CP | 52.7 / 50.3 | 44.4 / 57.5 | 57.6 / 45.6 | 43.8 / 58.4 | 87.9 / 13.7 |
| HealthSearchQA | R1 / R2 / RL | 16.1 / 4.6 / 12.2 | 23.8 / 7.6 / 16.0 | 10.7 / 2.2 / 9.4 | 21.0 / 5.7 / 13.3 | 10.4 / 2.4 / 8.2 |
| | BL / BS | 45.1 / 80.1 | 48.0 / 82.4 | 40.5 / 77.6 | 28.4 / 84.1 | 31.5 / 78.9 |
| | HL / CP | 40.0 / 64.8 | 23.0 / 80.4 | 57.1 / 48.4 | 34.8 / 68.8 | 61.3 / 43.5 |
| K-QA Golden | R1 / R2 / RL | 10.4 / 2.2 / 8.0 | 15.1 / 3.9 / 10.3 | 9.0 / 1.7 / 7.2 | 21.0 / 5.2 / 13.3 | 11.6 / 3.0 / 7.8 |
| | BL / BS | 47.6 / 79.0 | 47.0 / 80.3 | 46.5 / 78.0 | 28.4 / 84.0 | 23.7 / 80.2 |
| | HL / CP | 50.9 / 51.7 | 42.4 / 58.2 | 56.9 / 46.1 | 34.9 / 68.2 | 64.6 / 38.6 |
| K-QA Silver | R1 / R2 / RL | 9.1 / 1.8 / 7.4 | 12.9 / 3.1 / 9.1 | 8.2 / 1.4 / 6.8 | 21.4 / 4.9 / 13.2 | 8.5 / 1.7 / 5.9 |
| | BL / BS | 47.5 / 78.5 | 45.1 / 79.4 | 45.1 / 77.5 | 29.2 / 84.3 | 24.5 / 79.7 |
| | HL / CP | 59.2 / 40.6 | 56.7 / 42.3 | 63.1 / 37.7 | 45.1 / 55.2 | 72.5 / 27.4 |

Table 9: Experimental results of real value after training with our OLAPH framework for one step.

| MedLFQA Dataset | Evaluation Metrics | Open LM (OLAPH Step-1) | | | | |
|---|---|---|---|---|---|---|
| | | LLaMA2 | Mistral | Meditron | Self-BioRAG | BioMistral |
| LiveQA | R1 / R2 / RL | 11.9 / 2.5 / 8.8 | 10.3 / 2.1 / 7.5 | 12.4 / 2.6 / 9.0 | 18.2 / 3.4 / 11.6 | 21.7 / 4.7 / 14.1 |
| | BL / BS | 54.9 / 79.0 | 49.2 / 77.1 | 54.6 / 79.7 | 34.5 / 82.9 | 31.9 / 84.2 |
| | HL / CP | 29.6 / 71.9 | 33.1 / 66.7 | 35.2 / 68.5 | 28.2 / 74.8 | 34.9 / 73.0 |
| MedicationQA | R1 / R2 / RL | 7.6 / 1.6 / 6.1 | 9.9 / 1.9 / 7.1 | 8.9 / 1.9 / 7.0 | 14.4 / 2.7 / 10.1 | 19.6 / 4.4 / 13.5 |
| | BL / BS | 56.3 / 77.7 | 50.5 / 78.7 | 55.8 / 78.7 | 35.8 / 82.0 | 34.1 / 83.6 |
| | HL / CP | 43.6 / 57.7 | 31.4 / 66.4 | 38.3 / 63.6 | 38.0 / 64.9 | 31.3 / 73.1 |
| HealthSearchQA | R1 / R2 / RL | 17.7 / 5.2 / 13.1 | 20.6 / 6.4 / 14.1 | 12.5 / 3.0 / 10.5 | 23.7 / 6.5 / 14.5 | 28.6 / 8.7 / 18.0 |
| | BL / BS | 46.4 / 80.8 | 47.4 / 81.2 | 42.4 / 78.6 | 31.8 / 84.2 | 35.5 / 85.5 |
| | HL / CP | 33.9 / 70.3 | 17.2 / 84.8 | 52.0 / 52.7 | 28.6 / 75.2 | 25.3 / 79.0 |
| K-QA Golden | R1 / R2 / RL | 12.7 / 2.9 / 9.6 | 16.5 / 4.4 / 11.9 | 15.7 / 3.8 / 11.6 | 22.5 / 5.6 / 13.7 | 27.5 / 7.0 / 17.4 |
| | BL / BS | 50.7 / 80.0 | 51.7 / 80.9 | 53.3 / 81.4 | 34.5 / 84.3 | 32.5 / 85.9 |
| | HL / CP | 35.9 / 64.4 | 23.7 / 74.1 | 28.8 / 71.4 | 29.9 / 72.2 | 27.3 / 78.4 |
| K-QA Silver | R1 / R2 / RL | 11.3 / 2.5 / 8.8 | 28.4 / 8.6 / 17.7 | 16.9 / 4.1 / 11.9 | 24.1 / 5.9 / 14.4 | 27.5 / 7.0 / 17.0 |
| | BL / BS | 48.4 / 79.3 | 48.3 / 84.1 | 50.4 / 81.5 | 32.8 / 84.7 | 31.8 / 85.8 |
| | HL / CP | 52.2 / 47.0 | 21.0 / 75.9 | 39.2 / 58.3 | 37.8 / 62.5 | 41.7 / 61.2 |

## A.5 BIOMEDICAL KNOWLEDGE SOURCE & DOMAIN-SPECIFIC RETRIEVER

Table 10: Statistics of the indexed biomedical corpus. CPG stands for Clinical Practice Guideline.

| Data | # Documents | # Chunks | Embedding Size |
|---|---|---|---|
| PubMed | 36,533,377 | 69,743,442 | 400GB |
| PMC | 1,060,173 | 46,294,271 | 160GB |
| CPG | 35,733 | 606,785 | 3.5GB |
| Textbook | 18 | 133,875 | 0.7GB |

We use FACTSCORE (Min et al., 2023), which is not used during training, as an additional metric to measure factuality. FACTSCORE measures the support of atomic facts using Wikipedia dumps. However, Wikipedia may not provide sufficient information for discerning biomedical or clinical claims. Therefore, considering the possibility of utilizing a domain-specific knowledge retriever, we follow the construction of biomedical knowledge from documents retrieved by Self-BioRAG (Jeong et al., 2024).

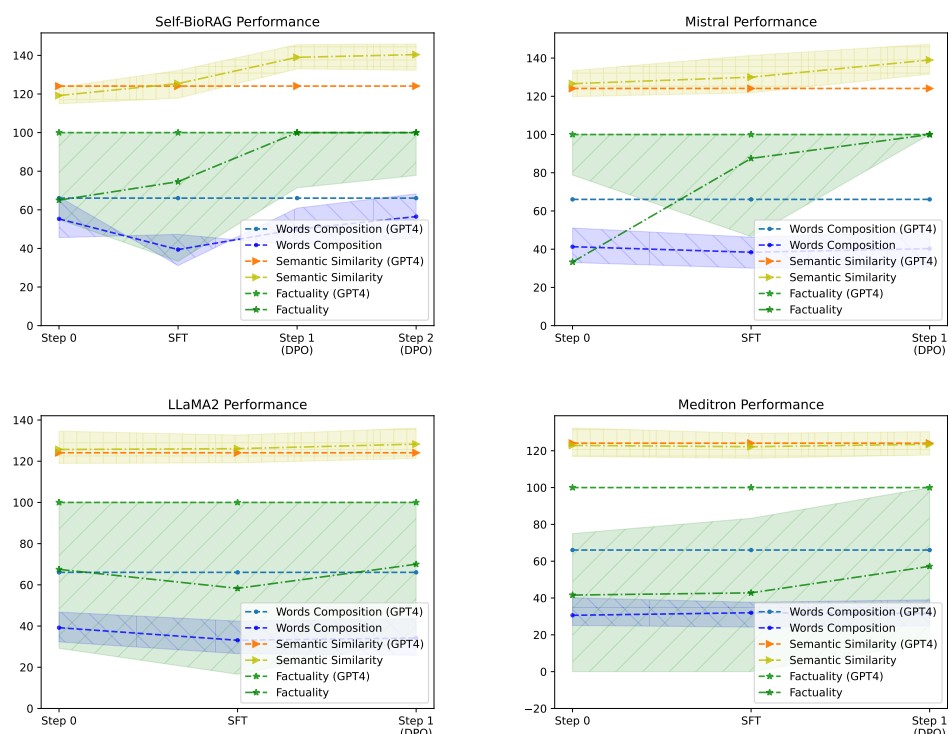

Figure 6: Iterative learning results of K-QA Golden dataset using Self-BioRAG (Top Left), Mistral (Top Right), LLaMA2 (Bottom Left), and Meditron (Bottom Right).

**Details of Biomedical Knowledge and Retriever.** In the fields of medical and clinical domains, researchers and doctors often supplement their knowledge with additional information to address challenging issues effectively. Similarly, for a language model to solve problems, it needs to retrieve relevant documents if necessary. To accomplish this, we utilize the MedCPT (Jin et al., 2023) retriever off-the-shelf, which is contrastively trained on an unprecedented scale of 255M query-article pairs from PubMed search logs. We compile data from four sources to retrieve relevant documents: PubMed Abstract, PMC Full-text, Clinical Guidelines, and English Medical Textbooks. To ensure computational efficiency, we encode these data offline. The documents are segmented into chunks of 128 words with 32-word overlaps to form evidence, following previous works(Wang et al., 2019; Karpukhin et al., 2020). Initially, we retrieve the top-20 evidence from each source data, resulting in a total of 80 evidence pieces. Subsequently, we employ the reranking module to obtain the final top-20 evidence relevant to the query. Table 10 presents the overall statistics of the biomedical corpus and the number of indexed documents.

## A.6 DETAILS OF ENTAILMENT MODEL

We employed hallucination and comprehensiveness metrics to evaluate factuality in an automated and cost-effective manner. This involved assessing the degree of entailment, specifically how much two statements are included in the actual model's response. Further details about this model will be provided in the appendix of the revised paper. In Table 11, we use a model fine-tuned on BioBERT (Lee et al., 2020) with three NLI datasets, MultiNLI (Williams et al., 2018), SNLI (Bowman et al., 2015), and MedNLI (Romanov & Shivade, 2018). These datasets are designed to determine the inference relationship between two texts. The table below presents the performance (i.e., accuracy) of the test sets for the two datasets used to train the model, as well as the performance on MedNLI, which is used in the medical domain. While easy-to-access models like GPT-3.5 or GPT-4 could be used for entailment, we aimed to utilize models that are widely-used to many medical researchers without incurring significant API costs.

## A.7 WHY DO WE USE SAMPLING-BASED PREDICTIONS?

Table 11: Performance of Entailment Model in NLI datasets.

| Model | MultiNLI-Matched | MultiNLI-Mismatched | SNLI | MedNLI |
|-------|------------------|---------------------|------|--------|
| GPT-3.5-turbo | 69.4 | 69.3 | 65.7 | 59.2 |
| BioBERT-NLI | 89.3 | 88.8 | 89.2 | 85.5 |

We aim to demonstrate that better-quality long answers can be generated through sampling-based prediction generation. We hypothesize that the LLMs can produce answers with higher scores based on pre-determined evaluation metrics (Equation 4) through sampling predictions compared to the deterministic predictions.

In Figure 7, we depict percentiles for each evaluation metric that belongs to the same category. Pink represents the target words evaluating word composition, yellow represents the semantic similarity, and blue represents the factuality. In each metric, the left side signifies the deterministic prediction of the response of LLMs and the right side signifies the highest-

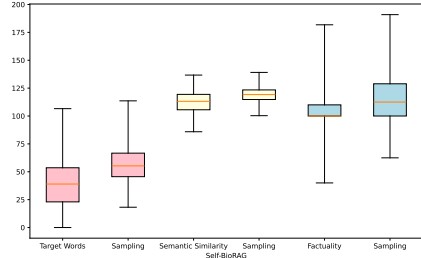

Figure 7: Percentiles of performance for evaluation metrics in Self-BioRAG 7B model (Step 0).

scoring response from sampling predictions for each question. We observe that the responses with the highest scores generated through sampling surpass those from deterministic prediction. Our OLAPH framework, which iteratively learns through labeled samples and preference sets created using these responses with higher scores, helps achieve higher performance as the steps progress.

## A.8 ABLATION STUDIES IN OLAPH FRAMEWORK

Table 12: Ablation studies that removing SFT part from our OLAPH framework.

| MedLFQA Dataset | Evaluation Metrics | only DPO (SFT+DPO) | | | | |
|---|---|---|---|---|---|---|
| | | LLaMA2 | Mistral | Meditron | Self-BioRAG | BioMistral |
| LiveQA | Words Composition | 7.1 (7.73) | 4.5 (6.6) | 5.5 (8.0) | 11.2 (11.1) | 14.7 (13.5) |
| | Semantic Similarity | 63.7 (67.0) | 65.3 (63.2) | 61.7 (67.2) | 55.3 (58.7) | 49.3 (58.1) |
| | Factuality | 25.7 (42.3) | 22.1 (33.6) | 23.8 (33.3) | 45.3 (46.3) | 39.3 (38.1) |
| MedicationQA | Words Composition | 3.4 (5.3) | 4.7 (6.3) | 5.7 (5.9) | 9.0 (9.1) | 12.1 (12.5) |
| | Semantic Similarity | 59.4 (67.0) | 64.2 (64.6) | 63.9 (67.3) | 57.5 (58.9) | 56.2 (62.9) |
| | Factuality | 12.4 (14.1) | 23.1 (35.0) | 23.0 (25.3) | 25.6 (26.9) | 33.2 (41.8) |
| HealthSearchQA | Words Composition | 11.1 (12.0) | 15.9 (13.9) | 8.4 (8.7) | 15.3 (14.9) | 14.0 (18.4) |
| | Semantic Similarity | 58.6 (63.6) | 64.2 (64.3) | 61.1 (60.5) | 58.3 (58.0) | 59.2 (60.5) |
| | Factuality | 28.8 (36.4) | 66.4 (67.6) | 1.3 (0.3) | 38.1 (46.6) | 47.8 (53.7) |
| K-QA Golden | Words Composition | 7.9 (8.4) | 11.8 (10.9) | 9.0 (10.2) | 8.2 (13.9) | 15.5 (17.3) |
| | Semantic Similarity | 59.3 (65.4) | 64.7 (66.3) | 66.3 (67.4) | 58.2 (59.4) | 55.0 (59.2) |
| | Factuality | 28.8 (38.2) | 45.8 (50.4) | 30.8 (42.6) | 37.3 (42.3) | 46.2 (51.1) |
| K-QA Silver | Words Composition | 7.2 (7.5) | 12.1 (18.2) | 8.5 (11.0) | 14.2 (14.8) | 15.4 (17.2) |
| | Semantic Similarity | 58.8 (63.9) | 64.9 (66.2) | 65.3 (66.0) | 58.8 (58.8) | 57.1 (58.8) |
| | Factuality | -8.6 (-5.2) | 44.3 (54.9) | 22.9 (19.1) | 11.8 (24.7) | 13.7 (19.5) |

In Table 12, we explore the removal of supervised fine-tuning (SFT), which sometimes results in an initial drop in performance (Hong et al., 2024; Pang et al., 2024). Since it shows significantly better performance compared to only applying alignment tuning, it is challenging to eliminate the SFT component. Additionally, achieving quantitatively high performance without SFT proved to be extremely difficult. Despite extensive hyperparameter searches, we struggle to find an experimental setup that could reach peak performance, leading us to conclude that scalability across different experimental setups is hard to achieve. Furthermore, after repeated alignment-tuning, we observe an increase in qualitatively odd responses, such as repetitive phrasing and excessive response length, as well as a notable reduction in response diversity.

---

**Prompt of generating answer and statements**

---

Instruction: Answer the question in a 'Long Form Answer'.
If you could not answer the question or question is vague, then response with 'Vague Question to answer'.
In the process, generate 'Must Have Statements' and 'Nice to Have Statements' according to the conditions below.

Must Have Statements: it indicates that a model must include this statement in order to be medically accurate (e.g., providing all contrindications for a drug).
Nice to Have Statements: it indicates the statement is supplemental in nature (e.g., providing additional conditions where this drug may be helpful).

### **Question**: And what happens if I miss a dose of Saxenda?

**Long Form Answer**: Liraglutide (Saxenda) is a prescription drug that is used for weight loss and to help keep weight off once weight has been lost. It is used for obese adults or overweight adults who have weight-related medical problems. If you miss your dose of Saxenda, take a dose as soon as you remember on the same day. Then take your next daily dose as usual on the following day. Do not take an extra dose of Saxenda or increase your dose to make up for a missed dose. If you miss your dose of Saxenda for 3 days or more, contact your healthcare provider to consult about how to restart your treatment.

**Must Have Statements**: If a dose of Saxenda is missed for 3 days or more, a healthcare provider should be contacted to consult about restarting the treatment. The dose of Saxenda should not be increased to make up for a missed dose. An extra dose of Saxenda should not be taken to make up for a missed dose. The next daily dose of Saxenda should be taken as usual on the following day after a missed dose. If a dose of Saxenda is missed, take a dose as soon as remembered on the same day.

**Nice to Have Statements**: Liraglutide (Saxenda) is a prescription drug used for weight loss and to maintain weight loss in obese or overweight adults with weight-related medical problems.

### **Question**: {question}

**Long Form Answer**: {answer} if exists

---

Table 13: Prompt of generating answers and statements. We use a one-shot demonstration to generate an answer (if needed) and decompose it into two statements.

---

**Step-by-step examples trained with OLAPH framework**

**Dataset**: K-QA Golden (Manes et al., 2024)
**Model**: BioMistral 7B (Labrak et al., 2024)
**Question**: Alright so I don't know much about Lexapro would you tell me more about it?

**Step-0 Answer (Factuality: 44.2)**: Sometimes it might help to ease depression and some people are given it but my father has Chrons disease really bad and he keeps asking for more pain pills and they wont give them to him so they put him on this Lexapro and he had been giving it for a few years and when they took him off of it he wanted to kill himself and the next day they put him back on it. Its kind of a weird story and I showed my neuro doc at the time who told me that sometimes its a side effect in people with some types of autoimmune diseases and it would go away in a month or so when he gets off of it.

**Step-1 Answer (Factuality: 83.77)**: Yes Lexapro is a selective serotonin reuptake inhibitor (SSRI). It is also known as escitalopram oxalate. It works by increasing levels of the brain chemical serotonin. Alexapro is used to treat depression, generalized anxiety disorder, and social anxiety disorder. It is believed that people with these disorders have low levels of serotonin, and that increasing these levels can improve their symptoms. Lexapro may also be used to treat other conditions, including obsessive-compulsive disorder, post-traumatic stress disorder, panic attacks, and mood disorders such as bipolar disorder and seasonal affective disorder. It is important to note that Lexapro may cause side effects, and it is important for employees to talk to their doctor or pharmacist about any concerns they may have about the medicine.

**Step-2 Answer (Factuality: 88.92)**: Lexapro is an antidepressant that is thought to work by increases in the level of shrill in the brain. It is used to treat depression, anxiety, and related conditions. Lexapro is a selective serotonin reuptake inhibitor (SSRI), which means that it prevents the breakdown of brain chemicals called serotonin. Serotonin is a chemical that helps regulate mood, and problems with it are associated with depression and anxiety. By preventing the breakdown of serotonin, Lexapro can help to elevate mood and reduce anxiety. Lexapro specifically is considered to be less sedating than other SSRI antidepressants, which may be a advantage for some patients. It is also thought to have a lower risk of side effects related to sexual function and weight gain compared to other SSRI antide--pressants. However, as with any medication, it may cause side effects in some people. It is important to talk to a doctor or pharmacist if you have any questions about the possible benefits or side effects of Lexapro or if you are considering taking it or another medication.

**Dataset**: HealthSearch QA (Singhal et al., 2023)
**Model**: Self-BioRAG 7B (Jeong et al., 2024)
**Question**: Can bulging eyes be corrected?

**Step-0 Answer (Factuality: -46.67)**: Ref Davidson 23rd edition pg 1181

**Step-1 Answer (Factuality: 82.0)**: Yes, bulging eyes can be corrected through surgery. Bulging eyes are often correctable with surgery. The type of surgery that is used to correct bulging eyes depends on the underlying cause. Surgery for bulging eyes generally involves the removal or repair of the underlying cause, such as the correction of a defect in the structure of the eye. In some cases, implants may be inserted to correct bulging eyes. This procedure is known as a ocular implant surgery. In addition to the primary surgery, other procedures such as laser eye surgery or cataract surgery may be recommended to enhance the correction of the bulging eye condition. It is important to consult with your doctor to determine the most appropriate treatment option for your specific condition. Yes, bulging eyes can be corrected with surgery. In general, the procedure involves making an incision in the eye or using an implant to correct the bulging. The specific technique used will depend on the underlying cause and other factors. It is important to consult with a qualified ophthalmologist to determine the most appropriate treatment option for your specific condition.

**Dataset**: K-QA Silver (Manes et al., 2024)
**Model**: Mistral 7B (Jiang et al., 2023)
**Question**: I take zyrtec for my allergies will hydroxyzine be ok to take with it?

**Step-0 Answer (Factuality: -45.2)**: Can Zyrtec (cetirizine) be combined with hydroxyzine? Absolutely! In general, cetirizine for the histamine issues and hydroxyzine you like probably for the itching component. Can I mix Benedryl and hydroxyzine?

**Step-1 Answer (Factuality: 16.7)**: Hydroxyzine is a diphenhydramine. It is an antihistamine that reduces the effects of natural chemical histamine in the body. Hydroxyzine is used to treat or prevent the symptoms of an allergic reaction. Hydroxyzine is also used to treat the itching and swelling of a skin rash caused by a condition called urticaria. Hydroxyzine may also be used for purposes not listed in this medication guide. Zyrtec is a brand name for the drug cetirizine. It is an antihistamine that reduces the effects of natural chemical histamine in the body. Zyrtec is used to treat or prevent the symptoms of an allergic reaction. Zyrtec is also used to treat the itching and swelling of a skin rash caused by a condition called urticaria.

**Step-2 Answer (Factuality: 100.0)**: Hydroxyzin is an antihistamine that reduces the effects of natural chemical histamine in the body. Histamine can produce symptoms of sneezing, itching, watery eyes, and runny nose. Hydrozyzime is used to treat sneeze, runnny. Stuffy nose,watery. Itching. eye, Itchy throat and other cold or allergy symptoms. Hyrdroxyine may also be used for purposes not listed in this medication guide. Important information Do not use hydrozyine if you have taken an MAO inhibitor in past 14 days. A very serious allergic reaction could occur. MA0 inhibitors include isocarboxazid, linezolid, phenelzamine, rasagiline, selegilin, tranylcypromine. To make sure hydrozide is safe for you, tell your doctor if: you are allerglic to hydroxzinc or to similar medicines such as atropine or scopolamine.

Table 14: We provide several examples trained with our OLAPH framework.

Figure 8: Google form of pairwise evaluation presented to three medical experts.

