# OpenReview forum: "OLAPH: Improving Factuality in Biomedical Long-form Question Answering"
_ICLR.cc/2025/Conference — Submitted to ICLR 2025_

### Official Review · Reviewer_DEzX · 2024-10-31

**Soundness:** 3
**Presentation:** 3
**Contribution:** 2
**Rating:** 6
**Confidence:** 4

**Summary:**

This research paper introduces OLAPH, a framework designed to improve the factual accuracy of long-form question answering in the medical domain. The authors address the challenge of hallucinations in large language models (LLMs) by utilizing a cost-effective and multifaceted automatic evaluation system to generate synthetic preference sets. These sets guide the LLMs in prioritizing factuality, semantic similarity, and word composition.

To facilitate automatic evaluation, the authors introduce MedLFQA, a reconstructed benchmark dataset that includes long-form answers and crucial statements derived from existing biomedical question-answering datasets. Through experiments, the authors demonstrate that even 7B LLMs, trained with OLAPH, can generate answers comparable to GPT-4 and medical experts in terms of factuality.

**Strengths:**

1. the data set collection follows good practice with human verification (high agreement).
2. method is clearly explained.
3. experiments are thorough (though I have questions and suggestions below).
4. most part of the paper is well written and easy to follow.

**Weaknesses:**

1. The method itself (OLAPH) is not really "novel" as the authors claim - the general frameworks has been used in many post-training of LLMs such as Llama and Claude models. I suggest softer contribution claims here.
2. Some claims are not very rigorous - see my questions below.
3. The final analysis can be made clearer.

Recommended related work.
RAG-QA Arena: Evaluating Domain Robustness for Long-form Retrieval Augmented Question Answering.

**Questions:**

1. For the analysis of or RQ 3, what's exactly experimental setup? I am not getting why you will need to supply domain knowledge here, and since you train multiple models in RQ 1 and 2, which models did you end up using for this?
2. In Figure 5, if using SFT leads to initial drop of performances, have you try to remove it? And what would the impact be?
3. Line 235 - 236, this may simply suggest that answers in K-QA is terrible, not necessarily that GPT-4 answers are good. Do you have more analysis here?
4. Line 248 - 252, I am not 100% convinced by this claim. Could it be that annotators are biased to believe whatever the model generates?
5. Line 282 + 283, do you use statements as SFT targets too? If so, what's the data format?

---

> ### Author Response · Authors · 2024-11-19
>
> We would like to kindly appreciate all the constructive and beneficial feedback and questions provided by the reviewers. We will provide comments addressing the weaknesses and questions raised by the reviewers.
>
> > The method itself (OLAPH) is not really "novel" as the authors claim - the general frameworks has been used in many post-training of LLMs such as Llama and Claude models. I suggest softer contribution claims here.
>
> The expressions have been toned down to some extent and reflected in the paper.
>
> > For the analysis of or RQ 3, what's exactly experimental setup? I am not getting why you will need to supply domain knowledge here, and since you train multiple models in RQ 1 and 2, which models did you end up using for this?
>
> As shown in lines 483-485 and Figure 5, the experiment measures the FactScore of Mistral, Self-BioRAG, and BioMistral trained with our OLAPH framework and also we set the GPT-4 and human expert scores from the K-QA golden dataset as upper bounds. The additional construction of domain knowledge (in here biomedical knowledge) to evaluate FactScore is due to the method's approach of dividing the model's response into atomic facts and comparing them with knowledge sources to measure entailment. We considered expanding the knowledge sources beyond the Wikipedia dump to include rich and diverse medical knowledge from resources such as PubMed, PMC full-texts, Clinical Practice Guidelines (CPG), and medical textbooks.
>
> As seen in Figure 5, when comparing the two lines representing the upper bound, it is evident that the biomedical knowledge sources slightly outperform the Wikipedia dump in FactScore. This indicates that even when incorporating domain-specific knowledge sources for additional comparison, our OLAPH framework still demonstrates meaningful improvements in factuality.
>
> > In Figure 5, if using SFT leads to initial drop of performances, have you try to remove it? And what would the impact be?
>
> Thank you for asking the ablation studies of our OLAPH framework. We've also explore the removal of SFT which sometimes results in an initial drop in performance. Since it shows significantly better performance compared to only applying alignment tuning, it is challenging to eliminate the SFT component. Additionally, achieving quantitatively high performance without SFT proved to be extremely difficult. Despite extensive hyper-parameter searches, we struggle to find an experimental setup that could reach peak performance, leading us to conclude that scalability across different experimental setups is hard to achieve.
> Furthermore, after repeated alignment-tuning, we observe an increase in qualitatively odd responses, such as repetitive phrasing and excessive response length, as well as a notable reduction in response diversity.
> We add this context into our appendix part.
>
> > Line 235 - 236, this may simply suggest that answers in K-QA is terrible, not necessarily that GPT-4 answers are good. Do you have more analysis here?
>
> According to the K-QA paper, since the K-QA answers were manually annotated by domain experts, it was difficult to question the quality of these answers. Therefore, we proceeded with the assumption that we only needed to verify with a medical expert whether the GPT-4 answers were better or not.
>
> > Line 248 - 252, I am not 100% convinced by this claim. Could it be that annotators are biased to believe whatever the model generates?
>
> As the reviewer mentioned, I agree that annotators could introduce bias when evaluating the model-generated statements. However, as authors with relatively limited domain knowledge in comparison to the six panel medical experts who validated the process, we did not perceive it as necessary to further re-validate the results on our own.
>
> > Line 282 + 283, do you use statements as SFT targets too? If so, what's the data format?
>
> For SFT, only the response with the highest automatic evaluation score among the generated sample answers for each model was used for training. Therefore, MH and NH statements were not utilized in the training process.
>
> We truly appreciate your thoughtful consideration and the time you've taken to engage with our work.

---

> > ### Comment · Reviewer_DEzX · 2024-11-22
> > **Reply to the authors**
> >
> > Thanks for your responses. I think my ratings remain appropriate.

---

### Official Review · Reviewer_oqAV · 2024-10-31

**Soundness:** 4
**Presentation:** 4
**Contribution:** 3
**Rating:** 8
**Confidence:** 3

**Summary:**

the paper introduces MedLFQA dataset to evaluate long-form answers in biomedical contexts. The authors generated this dataset by combining multiple existing datasets to enable comprehensive assessment. This fills a gap in the automated evaluation of LFQA in a domain where factual accuracy is critical. Also, the authors propose & evaluate OLAPH framework on word composition, semantic similarity, factuality- using benchmarks and metrics like FACTSCORE. This framework shows improvements (in factual accuracy) for 7B small parameter models. This study is a novel attempt to reduce hallucinations in medical responses.

**Strengths:**

The paper shows an innovative approach to enhancing the factual accuracy of long-form biomedical question answering. This work also creatively combines current techniques in preference-based learning and factual consistency checks to improve medical domain answers, addressing key limitations in factuality and response quality in prior research. This work is good in quality of methodological design. They have detailed each step in OLAPH’s alignment process making it easy for others to build on top of it. The evaluation approach is also comprehensive as they used well known metrics. The paper is clearly written and answers major questions readers (at least I) may have. The significance of OLAPH lies in its potential to impact the application of LLM in healthcare. Medlfqa dataset can help other researchers in developing and evaluating factually accurate medical response systems.

**Weaknesses:**

The framework instroduced in this study is using GPT-4 to generate must-have and nice-to-have statements in medlfqa. My concern is that it may introduce biases or inaccuracies into the dataset. Although the researchers show that GPT-4-generated responses are close to human-curated answers, i did not find a critical analysis of where synthetic statements might diverge from medical experts.

Also, i think the study framework is a good step towards medical LLMs, but it does not cover the framework's behavior for a domain-specific case like disease diagnosis or recommendation of treatment.

**Questions:**

1. Were there specific medical domains or types of questions where synthetic responses diverged from expert standards?
I think it would be helpful if you could discuss any plans to validate the quality of synthetic data using a diverse set of generative models to assess robustness.

2. As we know, medical knowledge advances over time. So, I am curious to understand how you plan to deal with the challenges of maintaining factual relevance if your framework/dataset is deployed over a longer period?

---

> ### Author Response · Authors · 2024-11-19
>
> First, thank you for recognizing the value of our paper. We deeply appreciate your understanding of the unique strengths we aimed to highlight, and we are grateful for your thoughtful feedback. Below, we provide answers to the points you raised.
>
> > As we know, medical knowledge advances over time. So, I am curious to understand how you plan to deal with the challenges of maintaining factual relevance if your framework/dataset is deployed over a longer period?
>
> This is an excellent question. Medical knowledge is continuously evolving and updating, and it has become a significant area of growth. As part of our future research, we are exploring the creation of chronological knowledge specifically for the medical domain to assess the extent of knowledge elicitation in existing open-source models. To maintain factual relevance, we are working on establishing a dataset construction pipeline that can continuously update changing knowledge. This would allow us to build a robust answer update pipeline, which could ultimately evolve into a comprehensive framework. While this idea is still in its early, somewhat naive stage with initial experiments underway, we believe it could become a crucial source of information in the medical field as it develops further.
>
> > it does not cover the framework's behavior for a domain-specific case like disease diagnosis or recommendation of treatment.
>
> The authors are currently conducting further research on a domain-specific case in history taking, aiming to develop a conversational agent that provides personalized medical diagnoses based on patient characteristics. The importance of factuality in this process is recognized, and exploring how the OLAPH framework can be applied to disease diagnosis would make for an excellent research direction.
>
> > Were there specific medical domains or types of questions where synthetic responses diverged from expert standards? I think it would be helpful if you could discuss any plans to validate the quality of synthetic data using a diverse set of generative models to assess robustness.
>
> I fully agree with the point raised. To address this, we plan to conduct a qualitative analysis using randomly sampled data instances from the LLaMA-3.1-70B model to compare it with the decomposed must-have and nice-to-have statements provided by medical experts. We will examine the differences in detail and update the paper accordingly. This will be done as soon as possible, but since it requires significant time and resources, we aim to provide a thorough analysis rather than a simple observation. We hope for your understanding in this regard.
>
> Once again, thank you sincerely for reviewing our paper and providing such valuable feedback. We truly appreciate your thoughtful consideration and the time you've taken to engage with our work.

---

> > ### Comment · Reviewer_oqAV · 2024-11-26
> >
> > Thank you so much for sharing your future research plans with me. Since most of my questions were for future research directions (and applicability of this work), I am going to keep my score intact.

---

### Official Review · Reviewer_Au2s · 2024-11-04

**Soundness:** 3
**Presentation:** 3
**Contribution:** 3
**Rating:** 6
**Confidence:** 4

**Summary:**

This paper aims to improve factual accuracy of large language models (LLMs) in generating long-form answers within the biomedical domain through two main contributions.
The authors first introduce MedLFQA, a benchmark dataset designed to facilitate the automatic evaluation of factual claims in LLM responses.
Next, the paper proposes OLAPH (Optimizing Large language models' Answers with Preferences of mitigating Hallucination) framework, which iteratively trains LLMs to reduce hallucinations and incorporate essential medical information by leveraging synthetic preference sets derived from cost-effective, multifaceted automatic evaluations.
Experimental results demonstrate that a 7B parameter LLMs trained with OLAPH can generate responses with improved factuality comparable to those of medical experts.

**Strengths:**

+  The paper addresses an important problem in the medical domain, where factuality is crucial for patient safety and trust in medical AI systems.
+ The paper is clearly written with well-structured presentation, clear visualizations, and illustrative examples.
+  The introduction of MedLFQA as a unified benchmark for evaluating factuality in biomedical LFQA is a valuable contribution to the field.
+ The effectiveness of OLAPH is comprehensively validated with thorough analyses, comparisons with proprietary models, and evaluation using metrics independent of the training process,

**Weaknesses:**

+ The novelty of the proposed OLAPH framework is limited, as it mostly follows the standard preference optimization process (SFT and DPO)

**Questions:**

+ The paper should address the convergence properties of OLAPH: Is convergence guaranteed? How do the number and types of evaluation metrics influence convergence rates?
+ Can the evaluation criteria for the qualification of generated data in section 3.2 be used as preference criteria for preference optimization process in section 4.2?
+ Why does the paper follow cross-validation approach for experiments instead of the traditional train/test split approach?

---

> ### Author Response · Authors · 2024-11-19
>
> First and foremost, we would like to appreciate all the constructive and beneficial feedback and questions provided by the reviewers.
> We will provide comments addressing the weaknesses and questions raised by the reviewers.
>
> > The novelty of the proposed OLAPH framework is limited, as it mostly follows the standard preference optimization process (SFT and DPO)
>
> We understand that the OLAPH approach may seem somewhat limited in novelty, given the presence of existing studies that iteratively apply SFT (Supervised Fine-Tuning) and DPO (Direct Preference Optimization) with rejection sampling or other sampling methods. However, we hope the novelty of our work is recognized in terms of how we design a meaningful synthetic dataset—whether for supervised learning or alignment tuning—and the direction of exploiting automatic evaluation, training details, and effectiveness of the expected outcomes when employing iterative training with under 7B size of models in the medical domain.
>
> > The paper should address the convergence properties of OLAPH: Is convergence guaranteed? How do the number and types of evaluation metrics influence convergence rates?
>
> By comparing Figure 4 and 6, it becomes evident that the guarantee of convergence varies depending on the evaluation metric. Specifically, this variation is particularly noticeable across different metrics. For instance, metrics like word composition and semantic similarity tend to show relatively minor differences between generated responses across samples. These differences are somewhat normalized, maintaining a consistent range or converging over iterations.
>
> However, for metrics evaluating factuality, such as hallucination and comprehensiveness, the variation between responses is significantly larger across samples. While these variations generally decrease as iterations progress, the extent of this reduction varies across models (particularly for LLaMA2 and Meditron). Thus, it cannot be confidently concluded that the results always converge.
> We plan to explore future research directions aimed at confidently ensuring convergence under these conditions.
>
> > Can the evaluation criteria for the qualification of generated data in section 3.2 be used as preference criteria for preference optimization process in section 4.2?
>
> This is a very intuitive question! The nine evaluation criteria utilized in Section 3.2 are assessed based on the definitions provided in Table 4, which serve as the foundation for evaluating QA pairs. To effectively apply these evaluations in Section 4.2, two prerequisites seem necessary:
> 1) A method for numerically representing each evaluation criterion, such as using rubrics for scoring.
> 2) A model capable of learning this process in a manner that minimizes error propagation.
>
> If the model’s response to a given query incorporates scores for each criterion—beyond our proposed cost-effective and multifaceted automatic evaluation—and normalizes them within the same range, it could significantly enhance the sophistication of the automatic evaluation process.
>
> > Why does the paper follow cross-validation approach for experiments instead of the traditional train/test split approach?
>
> As mentioned in the evaluation of the MedLFQA benchmark paragraph in Section 5, data is extremely scarce due to the unique nature of the medical domain and there is no dedicated training dataset for the entirety of MedLFQA. Therefore, we adopted a cross-validation approach, where one subset of the data was treated as the test dataset while the remaining subsets were used as the training dataset for learning and evaluation.
>
> We truly appreciate your thoughtful consideration and the time you've taken to engage with our work.

---

### Official Review · Reviewer_UBp6 · 2024-11-05

**Soundness:** 3
**Presentation:** 3
**Contribution:** 3
**Rating:** 5
**Confidence:** 3

**Summary:**

This paper proposed a dataset MedLFQA for automatic evaluation of factuality long-form question-answering in the biomedical domain. The authors also proposed a training framework and claimed that this framework can optimize LLMs to reduce hallucination step-by-step.
The study topic in this paper is important as the hallucination problem is critical when applying LLMs in the health domain. The manual evaluation and the dataset would be beneficial to the community.
My main concerns are:
1. The MedLFQA sets the answers in MUST HAVE and NICE TO HAVE and then calculates the hallucination and comprehensiveness metrics by comparing the generated text and the reference text. Although automatic hallucination detection and quantization are difficult, and it is worth exploring automatic evaluation methods, it is not persuasive that the current setting can effectively serve as the hallucination metric. The problems are: a) this setting only evaluates a subset of hallucination; the LLM generated incorrect facts that outside of the MH and NH will not be taken into account in the metrics; b) The calculation of contradicts and entails are based on a fine-tuned BioBERT model, which brings uncertainty of the evaluation, especially on the medical domain. The metrics are highly affected by the performance of the fine-tuned BioBERT model, so a sensitivity analysis needs to be tested. In total, it is not persuasive that the current hallucination and comprehensiveness metrics are sufficient.
2. It is no surprise that the proposed OLAPH framework can improve the three metrics (Word composition, Semantic Similarity, and Factuality), as these three metrics are directly optimized during the training (similar to the comparison that fine-tuned model is better than non fine-tuned).
3. Question: In Table 5, when setting the alpha_3 as 1.0, the performance seems better than other smaller alpha_3 scores. So if we enlarge the alpha_3 over 1 (2,3,5 or more), will the model's performance continue to improve? Furthermore, it would be interesting to see the performance of the single loss function (i.e., set two other alpha as zero) to confirm which loss function contributes the most.

**Strengths:**

The study topic in this paper is important as the hallucination problem is critical when applying LLMs in the health domain. The manual evaluation and the dataset would be beneficial to the community.

**Weaknesses:**

1. The MedLFQA sets the answers in MUST HAVE and NICE TO HAVE and then calculates the hallucination and comprehensiveness metrics by comparing the generated text and the reference text. Although automatic hallucination detection and quantization are difficult, and it is worth exploring automatic evaluation methods, it is not persuasive that the current setting can effectively serve as the hallucination metric. The problems are: a) this setting only evaluates a subset of hallucination; the LLM generated incorrect facts that outside of the MH and NH will not be taken into account in the metrics; b) The calculation of contradicts and entails are based on a fine-tuned BioBERT model, which brings uncertainty of the evaluation, especially on the medical domain. The metrics are highly affected by the performance of the fine-tuned BioBERT model, so a sensitivity analysis (with different models other than fine-tuned BioBERT) needs to be tested. In total, it is not persuasive that the current hallucination and comprehensiveness metrics are sufficient. These evaluation limitations need to be discussed.
2. It is no surprise that the proposed OLAPH framework can improve the three metrics (Word composition, Semantic Similarity, and Factuality), as these three metrics are directly optimized during the training (similar to the comparison that fine-tuned model is better than non fine-tuned).

**Questions:**

Question: In table 5, when set the alpha_3 as 1.0, the performance seems better that other smaller alpha_3 scores. Why not to enlarge the alpha_3 over 1 (2,3,5 or more) to test the model performance? Furthermore, it would be interesting to see the performance of the single loss function (i.e. set two other alpha as zero)

---

> ### Author Response · Authors · 2024-11-19
>
> First and foremost, we would like to appreciate all the constructive and beneficial feedback and questions provided by the reviewers. We will provide comments addressing the weaknesses and questions raised by the reviewers.
>
> > it is not persuasive that the current setting can effectively serve as the hallucination metric. This setting only evaluates a subset of hallucination; the LLM generated incorrect facts that outside of the MH and NH will not be taken into account in the metrics
>
> The authors truly agree that our suggested OLAPH framework evaluates the subset of hallucinations. By categorizing hallucinations into intrinsic and extrinsic types, our study focuses more on handling extrinsic hallucinations. Extrinsic hallucinations occur when the model generates information that is factually incorrect or contradicts external, verifiable facts. However, we believe that MH and NH can effectively measure against external and verifiable facts. This is because our approach trains the model to generate preferred answers which are decomposed into MH and NH statements and has shown improvements in experiments involving such facts.
>
> Additionally, when applying FactScore to retrieve relevant documents from resources like the Wikipedia dump and biomedical dump, and comparing them at the atomic fact level, we observed that the application of our OLAPH framework resulted in an increase in the FactScore. This suggests that measuring factuality by considering MH and NH positively influences external and verifiable facts.
>
> > The calculation of contradicts and entails are based on a fine-tuned BioBERT model, which brings uncertainty of the evaluation, especially on the medical domain. The metrics are highly affected by the performance of the fine-tuned BioBERT model, so a sensitivity analysis (with different models other than fine-tuned BioBERT) needs to be tested.
>
> The reviewer raises a very insightful point. I agree with this perspective, as using a single classification model to evaluate entailment can be incomplete (and potentially not robust), which may also introduce additional error propagation at the evaluation stage. However, as shown in Table 11, we proved that the fine-tuned BioBERT is sufficiently capable of measuring entailment as an NLI model. Still, we agree that reducing the uncertainty that may arise during the evaluation stage is a highly desirable direction. If the reviewer could provide an alternative NLI models that can measure entailment, we would be happy to conduct additional experiments and evaluate it using that model.
>
> > It is no surprise that the proposed OLAPH framework can improve the three metrics (Word composition, Semantic Similarity, and Factuality), as these three metrics are directly optimized during the training (similar to the comparison that fine-tuned model is better than non fine-tuned).
>
> That's an excellent point. The three evaluation categories reviewer mentioned should indeed be directly optimized to improve performance. However, since our experimental setup ensures a clear separation between the training and test sets—where the model is trained solely on the training set without exposure to the test set—it seems that this does not pose a significant issue.
>
> > In table 5, when set the alpha_3 as 1.0, the performance seems better that other smaller alpha_3 scores. Why not to enlarge the alpha_3 over 1 (2,3,5 or more) to test the model performance?
>
> In lines 831-857, alpha was utilized as a means to normalize the range of the multifaceted automatic evaluation from the perspective of convex optimization. Therefore, we did not consider experimenting with allocating a higher value to alpha_3. However, based on the results in Table 6, it seems possible to predict the outcomes of experiments where the values of alpha_1 and alpha_2 are set to 0. In Table 6, experiments were conducted with alpha_3 set to a value between 0 and 1. Although alpha_1 and alpha_2 did not show as significant performance changes as alpha_3, we observed that learning to decompose MH and NH for entailment led to improvements in word composition and semantic similarity. Therefore, it can be anticipated that if we set alpha_1 and alpha_2 to 0 and gradually increase the value of alpha_3, the performance will align and show a slight improvement.
>
> The primary goal of the OLAPH framework is to integrate an automatic multifaceted evaluation into the learning process. The core idea is that this allows the model to generate synthetic data using self-generated responses for training. While the current metrics, which are highly correlated with each other, show an aligned performance trend as discussed above, there is a high likelihood that new evaluation metrics—such as fluency and sympathy—will yield different patterns when automatically incorporated. Keeping this in mind will provide a clearer perspective on the results.
>
> We truly appreciate your thoughtful consideration and the time you've taken to engage with our work.

---

### Author Response · Authors · 2024-11-27

Dear Area Chair and Reviewers,

We would like to extend our heartfelt thanks to you and the reviewers for your thoughtful and constructive feedback on our submission. Your comments and suggestions have been instrumental in helping us improve the clarity and quality of our work.

We deeply appreciate the significant time and effort that the reviewers dedicate to the review process. With only a few days remaining in the discussion phase, we are eager to engage further and address any remaining questions or concerns.

We fully understand the time constraints that reviewers face, but we sincerely hope they might have an opportunity to review our revisions before the discussion phase concludes.

Thank you once again for your time and kind consideration.

Sincerely,

The Authors

---

### Meta-Review · Area_Chair_nZfX · 2024-12-20

**Metareview:**

This work concerns using LLMs to answer clinical queries directly. The main contribution is the compilation of a meta-resource for medical question answering, which is a composition of existing corpora that the authors call MedLFQA. The authors then follow what appears to be a fairly standard training recipe (SFT followed by preference tuning via DPO), informed by a composite metric that combines multiple facets (Eq 4). They refer to this training approach as OLAPH, though I agree with reviewers DEzX and Au2s: This is not particularly novel, methodologically (the authors also seem to acknowledge this in their response).

I have two major concerns about this work. First, the framing. This paper is about "addressing patients’ questions", and takes for granted that we want to use LLMs to do this. Do we? Is adopting LLMs in this way worth the risk? Don't we want providers in the loop to inform patient care? No discussion of this is found in the paper; it is simply taken for granted that using LLMs to directly answer patient medical questions is a good idea. In the response period, the authors write that they are "aiming to develop a conversational agent that provides personalized medical diagnoses based on patient characteristics."; this is a very thorny proposition, given the risks, but again there is scant discussion of this in the paper. That said, I understand that there is precedent for this as a "task" (hence the availability of existing datasets that the authors compile into MedLFQA) and I also do appreciate that making technical progress on increasing factuality is a valid goal unto itself, and medical QA provides a compelling context for investigating this. So, to be clear, while I take issue with the framing, this would not be means on its own for rejecting the paper in my view.

Second, and more practically, I share the technical concerns raised by UBp6, and do not feel these were adequately addressed in response and revision. Specifically:

— I am not convinced that (Bio)BERT trained for NLI is likely to be reliable at for the inference task here. The authors point to Table 11 in the Appendix as "proof" of sufficiency, but this is not particularly compelling. This table reports results of GPT-3.5-Turbo and BioBERT *fine-tuned* apparently on MedNLI (the authors write "Further details about this model will be provided in the appendix of the revised paper" in the current draft, so I can only speculate). The latter does OK, the former does poorly. But of course the authors are using the fine-tuned model out of domain, so these numbers are extremely optimistic and not applicable to the samples under consideration. The authors acknowledge this as a weakness in their response, but there is no plan to address it.

— The evaluation is with respect to the same aspects that are used for the composite objective which informs preference tuning via DPO. Therefore, it is indeed unsurprising that "their" model "outperforms" other approaches: It was trained to optimize the scores being measured. The authors respond to this critique by saying "since our experimental setup ensures a clear separation between the training and test sets—where the model is trained solely on the training set without exposure to the test set—it seems that this does not pose a significant issue.". But this misses the point, which is that if you optimize a model for some measure and compare it to models that were not (which is the case here), you will be all but guaranteed that said model will "win"; this basically just tells us that the optimization was successful.

**Additional Comments On Reviewer Discussion:**

The authors did helpfully clarify some points in discussion, but major issues (discussed above) were inadequately addressed in my view.

---

### Decision · Program_Chairs · 2025-01-22

Reject